# From Samples to Scenarios: A New Paradigm for Probabilistic Forecasting

**Xilin Dai[1]*** **Zhijian Xu[2]*** **Wanxu Cai[3]** **Qiang Xu[2]†**

[1]ZJU-UIUC Institute, Zhejiang University
[2]Department of Computer Science and Engineering, The Chinese University of Hong Kong
[3]School of Software, Tsinghua University
`xilin2023@zju.edu.cn, zjxu21@cse.cuhk.edu.hk`
`caiwx22@mails.tsinghua.edu.cn, qxu@cse.cuhk.edu.hk`

## Abstract

Most state-of-the-art probabilistic time series forecasting models rely on sampling to represent future uncertainty. However, this paradigm suffers from inherent limitations, such as lacking explicit probabilities, inadequate coverage, and high computational costs. In this work, we introduce **Probabilistic Scenarios**, an alternative paradigm designed to address the limitations of sampling. It operates by directly producing a finite set of {Scenario, Probability} pairs, thus avoiding Monte Carlo-like approximation. To validate this paradigm, we propose **TimePrism**, a simple model composed of only three parallel linear layers. Surprisingly, TimePrism achieves 9 out of 10 state-of-the-art results across five benchmark datasets on two metrics. The effectiveness of our paradigm comes from a fundamental reframing of the learning objective. Instead of modeling an entire continuous probability space, the model learns to represent a set of plausible scenarios and corresponding probabilities. Our work demonstrates the potential of the Probabilistic Scenarios paradigm, opening a promising research direction in forecasting beyond sampling.

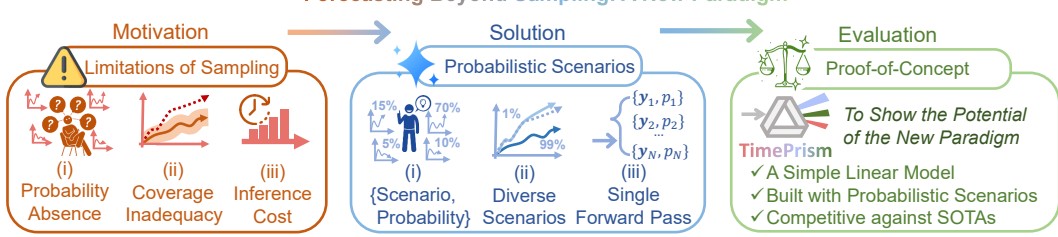

Figure 1: **Motivation, solution, and evaluation of this work.** We illustrate the limitations of the prevailing sampling-based paradigm for probabilistic forecasting. In response, we introduce Probabilistic Scenarios, a new paradigm that directly produces a set of {Scenario, Probability} pairs, and validate its potential with a simple proof-of-concept model, TimePrism.

## 1 Introduction

Probabilistic time series forecasting is fundamental to optimal decision-making under uncertainty, as it describes the likelihood of future outcomes (Gneiting & Katzfuss, 2014; Hyndman & Athanasopoulos, 2021). Although this problem has been studied extensively within the machine learning community, current approaches tend to rely on a predefined predictive distribution or sampling approximation (Kong et al., 2025; Fang & Wang, 2020; Lim & Zohren, 2021). These strategies have led to three main categories of models: *(i) Parametric Distribution Models*, assumes that the predictive distribution conforms to a predefined parametric family, such as a Gaussian (Salinas et al., 2020); *(ii) Generative Models*, such as diffusion-based models (Rasul et al., 2021), which learn an iterative process to

---

*Equal contribution.
†Corresponding author.

generate samples from the latent distribution without explicitly defining its density function; and *(iii) Structured Probabilistic Models*, such as Flow-based Models (Rasul et al., 2020; Ashok et al., 2023), which learns a continuous probability density field, from which trajectories are then sampled.

However, the reliance on sampling introduces challenges (Cortés et al., 2025). While the alternative of a predefined distribution is not widely discussed by state-of-the-art methods for its evident inflexibility (Zhang et al., 2024; Ashok et al., 2023), the sampling paradigm suffers from three primary limitations, as shown in Figure 1: **(i) Probability Absence**. The most significant issue is that the generated trajectories are not paired with their probability of occurrence. Although confidence intervals can be inferred from a large set of samples, this process is indirect, computationally intensive, and lacks the intuitiveness of a direct scenario-probability mapping. **(ii) Coverage Inadequacy**. A finite set of samples may fail to represent low-probability, high-impact tail events. This is a critical failure for applications where preparing for rare occurrences is paramount, such as extreme weather or stock market volatility (Cortés et al., 2025). **(iii) Inference Cost**. The process of generating multiple samples is often expensive, with costs scaling to the number of samples required. This expense exacerbates the two preceding issues in practical applications, limiting the reliability and utility of such forecasts (Chen & Boccelli, 2018; Ashok et al., 2023).

To address the limitations of sampling-based forecasting, we introduce a new paradigm for probabilistic forecasting that we term **Probabilistic Scenarios**. The objective is to produce, in a single forward pass and without reliance on sampling, a finite set of **{Future Scenario, Probability}** pairs that explicitly represents the predictive distribution. While some state-of-the-art deep learning models have made progress toward this goal, none have fully achieved it. Structured probabilistic models like TempFlow and TACTiS/TACTiS2 can compute probabilities, but only as a continuous density field (Rasul et al., 2020; Ashok et al., 2023; Drouin et al., 2022), not as discrete, interpretable scenarios; obtaining explicit trajectories still requires reverting to the expensive sampling. Meanwhile, although TimeMCL (Cortés et al., 2025) produces a set of discrete scenarios, its optimization objective prioritizes scenario fidelity over probability matching.

To realize and validate the concept of Probabilistic Scenarios, we designed a proof-of-concept model, **TimePrism**. As its name suggests, TimePrism processes the input history to generate a discrete set of distinct future trajectories, which we term Scenarios, and concurrently estimates their likelihood to yield a set of {Future Scenario, Probability} pairs. Specifically, to validate the effectiveness of our paradigm, the architecture is intentionally kept simple. TimePrism is composed of only three parallel linear layers, designed end-to-end to directly produce Probabilistic Scenarios. Despite its simplicity, TimePrism achieves 9 out of 10 state-of-the-art (SOTA) results and one second-best result across five benchmark datasets on our two primary metrics.

**Contributions**:

- We introduce a new paradigm for probabilistic time series forecasting. This paradigm addresses the limitations of sampling by reframing the learning objective from continuous probability space estimation to a more structured task of learning a distribution over a set of scenarios
- For quantitative measurements, we establish an evaluation framework with two complementary metrics and provide distinct but comparable formulations for both sampling-based models and our paradigm, serving as a fair standard for future research on Probabilistic Scenarios.
- To show the potential of our paradigm, we introduce TimePrism, a simple linear model built within the Probabilistic Scenarios paradigm. Despite its simple structure, TimePrism still achieves competitive performance against SOTA sampling-based models, indicating a promising research direction for forecasting with Probabilistic Scenarios.

## 2 RELATED WORKS

**Parametric Distribution Models** employ a neural network to output the parameters of a prespecified probability distribution (Wu et al., 2020), with examples including DeepAR (Salinas et al., 2020), GPVar (Salinas et al., 2019), and Mixture Density Networks (Li et al., 2024). The primary limitation of this approach is its reliance on a strong distributional assumption. Due to this inherent inflexibility, this approach is less commonly adopted in recent SOTA methods (Zhang et al., 2024).

**Generative Models** represent the predictive distribution implicitly through a learned sampling process (Ho et al., 2020). For instance, Rasul et al. (2021) (TimeGrad) and Alcaraz & Strodthoff

(2022) (SSSD) pioneered the use of conditional diffusion models for probabilistic forecasting and imputation. Recent advancements have focused on adapting the diffusion process to the sequential nature of time series (Gao et al., 2025; Biloš et al., 2023). Another direction focuses on enhancing the conditioning mechanism (Kollovieh et al., 2023; Liu et al., 2025) and adding cross-modal visual information (Ruan et al., 2025). Further adaptations include designing non-stationary processes (Ye et al., 2025) and specializing models for tasks like imputation, such as CSDI (Tashiro et al., 2021). Related work also employs Variational Autoencoders (VAEs), such as GP-VAE (Fortuin et al., 2020). While powerful, all these methods produce forecasts via an iterative sampling procedure and do not provide an explicit probability density for any given trajectory.

**Structured Probabilistic Models** are a class of methods that learn an explicit, continuous probability density field over the forecast horizon, primarily including flow-based models and copula-based models. Flow-based models learn a distribution transformation (Papamakarios et al., 2021), with recent applications in time series forecasting, such as TempFlow, using techniques like conditioned normalizing flows and flow matching (Rasul et al., 2020; Kollovieh et al., 2024). Copula-based models, which have a long history in econometrics and finance (Patton, 2012; Größer & Okhrin, 2022), construct a joint distribution by a copula (Salmon & Bouyé, 2008; Bouyé et al., 2008; Wang & Tao, 2020). With neural networks involved to model the copula (Wen & Torkkola, 2019; Krupskii & Joe, 2020; Mayer & Wied, 2023; Toubeau et al., 2019), recent research leads to fully neural, non-parametric approaches like TACTiS (Drouin et al., 2022) and its successor, TACTiS-2 (Ashok et al., 2023). Despite their different formulations, models in this category still rely on a sampling procedure, drawing from a continuous probability density field to obtain future trajectories.

**Multiple Choice Learning** (MCL) framework offers a practical path toward realizing our Probabilistic Scenarios paradigm (Cortés et al., 2025). MCL, introduced by Guzmán-rivera et al. (2012), uses a Winner-Takes-All (WTA) loss to train a multi-head network, where each head specializes in a different mode of the data. This approach has been successfully applied and extended in various domains, particularly computer vision and reinforcement learning (Lee et al., 2016; Rupprecht et al., 2017; Tian et al., 2019; Seo et al., 2020; Garcia et al., 2021). Recent work has further analyzed its geometric properties and variants (Letzelter et al., 2024; 2023; Perera et al., 2024). In probabilistic time series forecasting, TimeMCL (Cortés et al., 2025) produces a discrete set of scenarios. However, its score heads do not directly model a probability distribution. Consequently, to compute probabilistic metrics like Continuous Ranked Probability Score (CRPS), the original work resamples from its finite set of scenarios. The authors acknowledge their prioritization of scenario fidelity over probability matching. This design addresses **Coverage Inadequacy** to some extent, but fails to solve **Probability Absence**. As reported in their work, this trade-off results in CRPS scores less competitive than SOTA models such as TACTiS-2 (Ashok et al., 2023) and TimeGrad (Rasul et al., 2021).

To transcend this trade-off, our Probabilistic Scenarios paradigm unifies scenario fidelity and probability matching, designed to address all three limitations of sampling coherently: **Probability Absence**, **Coverage Inadequacy** and **Inference Cost**.

## 3 THE PROBABILISTIC SCENARIOS PARADIGM

### 3.1 CONVENTIONAL FORECASTING PARADIGM WITH SAMPLING

We begin by formalizing the objective of probabilistic time series forecasting. Given a historical context window of length $L$, denoted as $\mathbf{x} = (x_1, \ldots, x_L) \in \mathbb{R}^{L \times D}$, where $D$ is the number of variates, the goal is to predict the distribution of the future trajectory over a horizon $T$, denoted as $\mathbf{y} = (y_1, \ldots, y_T) \in \mathbb{R}^{T \times D}$. The objective is to learn a model that captures the conditional probability distribution over all possible future trajectories:

$$P(\mathbf{y}|\mathbf{x}) \tag{1}$$

Directly modeling this high-dimensional distribution is often intractable. Consequently, state-of-the-art sampling-based methods learn a model, parameterized by $\theta$, that represents this distribution, denoted as $P_\theta(\mathbf{y}|\mathbf{x})$. A single forecast sample, $\hat{\mathbf{y}}$, is generated by sampling from this learned distribution in equation 2. The final probabilistic forecast is then represented by a set of $S$ such samples, $\mathcal{Y}_{\text{samples}} = \{\hat{\mathbf{y}}_i\}_{i=1}^S$, where $S$ is the number of samples. This set serves as an empirical Monte Carlo approximation of the true conditional distribution in equation 1. This workflow leads to

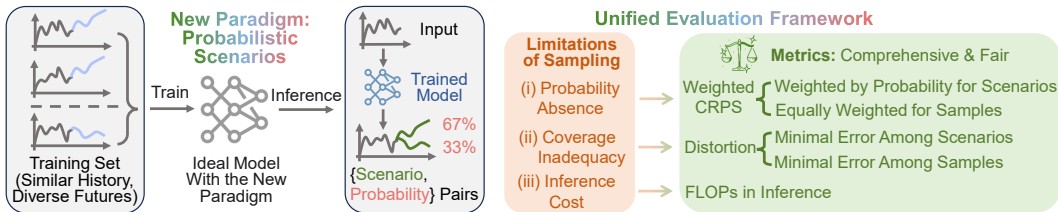

Figure 2: **Probabilistic Scenarios Paradigm and Unified Evaluation Framework.** The left panel illustrates an ideal behavior: a model trained on a dataset where similar histories lead to diverse futures should learn to output {Scenario, Probability} pairs that reflect the empirical frequency of those futures. The right panel details our evaluation framework, which links the limitations of sampling to adapted metrics and provides distinct yet comparable formulations for both paradigms.

the mentioned limitations: the **Probability Absence** for any given sample $\hat{\mathbf{y}}_i$, the risk of **Coverage Inadequacy** when the set $\mathcal{Y}_{\text{samples}}$ fails to capture rare but critical events, and the high **Inference Cost** of generating a sufficient number of samples.

$$\hat{\mathbf{y}} \sim P_\theta(\mathbf{y}|\mathbf{x}) \tag{2}$$

### 3.2 New Paradigm: Probabilistic Scenarios

In light of the challenges in sampling, we explore an alternative paradigm that reframes the forecasting objective, as illustrated in Figure 2. Instead of learning a generative process to approximate a continuous distribution, our paradigm aims to learn a direct mapping from the historical context to a discrete, finite probability space of future scenarios. Formally, we define a model under this paradigm as a function, $f$, that maps the history $\mathbf{x}$ to a tuple containing both the set of all future scenarios and their corresponding probabilities:

$$f(\mathbf{x}) = (\mathcal{Y}_{\text{pred}}, \mathbf{p}) \tag{3}$$

where: $\mathcal{Y}_{\text{pred}} = \{\mathbf{y}_n\}_{n=1}^N$ is the finite set of $N$ predicted future scenarios, with each scenario $\mathbf{y}_n \in \mathbb{R}^{T \times D}$. $\mathbf{p} = (p_1, \dots, p_N)$ is the vector of probabilities associated with the scenarios in $\mathcal{Y}_{\text{pred}}$. The probabilities must satisfy the axioms $p_n \geq 0$ for all $n$ and $\sum_{n=1}^N p_n = 1$.

This formulation directly yields an explicit set of **{Scenario, Probability}** pairs in a single forward pass. It differs from the Monte Carlo approximation of equation 2 by providing a discrete probability distribution that is both interpretable and computationally efficient. In essence, this paradigm neither assumes a parametric distributional form nor requires sampling, but instead learns to end-to-end generate {Scenario, Probability} pairs. This reframing of the objective simplifies the learning problem. A detailed discussion and theoretical analysis are provided in the Appendix A.1.

### 3.3 Unified Evaluation Framework

To quantitatively measure the limitations of sampling-based methods, we establish an evaluation framework by adapting two complementary metrics. We use the standard metric for overall forecast quality (Zhang et al., 2024; Zheng & Sun, 2025), the **CRPS**, to assess **Probability Absence**. Concurrently, we use **Distortion**, defined as the error of the best single trajectory in a set, to assess **Coverage Inadequacy** (Cortés et al., 2025). For both metrics, we provide distinct but comparable formulations for the sampling-based and Probabilistic Scenarios paradigms.

**Weighted CRPS for Probability Absence** We employ the energy score formulation of CRPS, which is defined for a single ground truth observation $\mathbf{y}_{\text{gt}}$ and a set of forecasts as $\mathbb{E}[\|\mathbf{y}-\mathbf{y}_{\text{gt}}\|] - \frac{1}{2}\mathbb{E}[\|\mathbf{y}-\mathbf{y}'\|]$, where $\mathbf{y}$ and $\mathbf{y}'$ are independent samples from the forecast distribution. We generalize this to our discrete, weighted scenario set. Given a set of $N$ scenarios $\mathcal{Y}_{\text{pred}} = \{\mathbf{y}_n\}_{n=1}^N$ and a corresponding probability vector $\mathbf{p} = (p_1, \dots, p_N)$, the Weighted CRPS is defined as:

$$\text{CRPS}(\mathcal{Y}_{\text{pred}}, \mathbf{p}, \mathbf{y}_{\text{gt}}) = \sum_{n=1}^N p_n \|\mathbf{y}_n - \mathbf{y}_{\text{gt}}\|_1 - \frac{1}{2}\sum_{n=1}^N \sum_{j=1}^N p_n p_j \|\mathbf{y}_n - \mathbf{y}_j\|_1 \tag{4}$$

where $\|\cdot\|_1$ denotes the L1 norm summed over all elements of the trajectory. We apply this formulation to both paradigms:

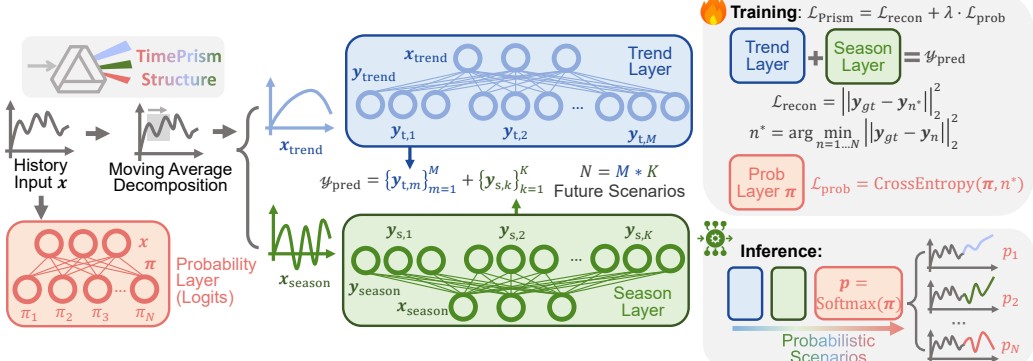

Figure 3: **Structure of TimePrism**, a linear model to demonstrate the potential of the Probabilistic Scenarios paradigm. The model operates in three parallel streams: after an initial decomposition, separate linear layers generate a basis of $M$ trend and $K$ seasonal forecasts. Simultaneously, a third linear layer produces the $N = M * K$ logits from the undecomposed history. This architecture, built within the Probabilistic Scenarios paradigm, achieves competitive performance despite its simplicity, demonstrating the potential of the new paradigm.

- **For Probabilistic Scenarios**, the scenarios $\{\mathbf{y}_n\}_{n=1}^N$ and probabilities $\{p_n\}_{n=1}^N$ are taken directly from the model's output $(\mathcal{Y}_{\text{pred}}, \mathbf{p})$.
- **For sampling-based models**, the evaluation is performed on the generated set of $S$ samples, $\mathcal{Y}_{\text{samples}} = \{\hat{\mathbf{y}}_i\}_{i=1}^S$. Each sample is assigned a uniform probability, i.e., $p_i = 1/S$.

The CRPS directly rewards models that assign higher probabilities to scenarios that are closer to the ground truth, thus quantitatively measuring the impact of *Probability Absence*.

**Distortion for Coverage Inadequacy.** Distortion measures the best-case performance of a forecast, quantifying how well the generated set of trajectories covers the true outcome (Cortés et al., 2025). It is defined as the minimum error between any single trajectory in the set and the ground truth. Following the implementation in our evaluation code, we define it as the minimum Root Mean Squared Error (RMSE) over the set of trajectories:

$$\text{Distortion}(\mathcal{Y}, \mathbf{y}_{\text{gt}}) = \min_{\mathbf{y}_n \in \mathcal{Y}} \sqrt{\frac{1}{T \cdot D} \left\| \mathbf{y}_n - \mathbf{y}_{\text{gt}} \right\|_F^2} \tag{5}$$

where $\| \cdot \|_F$ is the Frobenius norm. We apply this formulation as follows:

- **For Probabilistic Scenarios**, the minimization is performed over the complete set of $N$ scenarios generated by the model, $\mathcal{Y} = \mathcal{Y}_{\text{pred}}$.
- **For sampling-based models**, the minimization is performed over the set of $S$ generated samples, $\mathcal{Y} = \mathcal{Y}_{\text{samples}}$.

This metric directly assesses the diversity and reach of the generated set of futures. A lower Distortion score indicates better coverage, particularly for tail events that may be missed by a limited number of samples, thus measuring *Coverage Inadequacy*.

## 4 TIMEPRISM: A PROOF-OF-CONCEPT MODEL

### 4.1 DESIGN PHILOSOPHY

The primary goal of TimePrism is not to introduce a new complex architecture, but to serve as a clear proof-of-concept for the Probabilistic Scenarios paradigm. We intentionally adopt a minimalist design to test a core hypothesis: that the Probabilistic Scenarios paradigm can prove effective even when implemented with a simple model architecture.

To this end, we construct TimePrism using only three parallel linear layers as its core learnable components, devoid of any non-linear activation functions or deep, stacked layers. This deliberate

simplicity acts as a controlled experiment. By stripping away architectural complexity, we ensure that the model's strong performance can be directly attributed to the strengths of the paradigm itself.

## 4.2 TimePrism Architecture

The architecture of TimePrism is illustrated in Figure 3. It consists of three parallel streams that process the input history to generate the final set of Probabilistic Scenarios. Inspired by recent works such as DLinear and FITS (Zeng et al., 2023; Xu et al., 2023), which showed that simple architectures can effectively validate a new paradigm, we use a backbone based on decomposition and linear layers. The model operates as follows.

*1) Decomposition:* First, the input history $\mathbf{x} \in \mathbb{R}^{L \times D}$ is separated into a trend component $\mathbf{x}_{\text{trend}}$ and a seasonal component $\mathbf{x}_{\text{season}}$ using a moving average filter. This is a standard decomposition technique where:

$$\mathbf{x}_{\text{season}} = \mathbf{x} - \mathbf{x}_{\text{trend}}, \quad \text{with} \quad \mathbf{x}_{\text{trend}} = \text{AvgPool}(\text{Padding}(\mathbf{x})) \tag{6}$$

*2) Trend and Season Layers:* The two decomposed components are then fed into two independent linear layers. The trend layer maps the trend component $\mathbf{x}_{\text{trend}}$ to a set of $M$ distinct trend forecasts, $\mathcal{M} = \{\mathbf{y}_{\text{t},m}\}_{m=1}^{M}$. Concurrently, the season layer maps the seasonal component $\mathbf{x}_{\text{season}}$ to a set of $K$ distinct seasonal forecasts, $\mathcal{K} = \{\mathbf{y}_{\text{s},k}\}_{k=1}^{K}$. The complete set of $N = M \cdot K$ future scenarios, $\mathcal{Y}_{\text{pred}}$, is constructed by combining these two sets:

$$\mathcal{Y}_{\text{pred}} = \mathcal{M} + \mathcal{K} = \{\mathbf{y}_{\text{t},m} + \mathbf{y}_{\text{s},k} \mid \mathbf{y}_{\text{t},m} \in \mathcal{M}, \mathbf{y}_{\text{s},k} \in \mathcal{K}\} \tag{7}$$

*3) Probability Layer:* Operating in parallel to the scenario generation, a third linear layer acts as the probability module. This layer takes the original, undecomposed history $\mathbf{x}$ as input and directly produces a logits vector $\boldsymbol{\pi} \in \mathbb{R}^{N}$. Each element $\pi_n$ in this vector corresponds to one of the $N$ scenarios generated via the combinatorial process in Eq. equation 7.

## 4.3 Training and Inference

**Loss Function and Training.** The design of the loss function is directly guided by the Probabilistic Scenarios paradigm. Specifically, the loss function, $\mathcal{L}_{\text{Prism}}$, is composed of two terms, each designed to supervise one component of the target **{Scenario, Probability}** output. The reconstruction loss, $\mathcal{L}_{\text{recon}}$, is responsible for optimizing the fidelity of the generated Scenarios. Concurrently, the probability loss, $\mathcal{L}_{\text{prob}}$, supervises the learning of a meaningful probability distribution over these scenarios. The coefficient of the probability term, $\lambda$, is set to 1 in this work. Given the ground truth future trajectory $\mathbf{y}_{\text{gt}}$, the total loss is:

$$\mathcal{L}_{\text{Prism}} = \mathcal{L}_{\text{recon}} + \lambda \cdot \mathcal{L}_{\text{prob}} \tag{8}$$

*For Scenarios:* The reconstruction loss, $\mathcal{L}_{\text{recon}}$, uses the Winner-Takes-All (WTA) principle. It first identifies the index $n^*$ of the scenario in $\mathcal{Y}_{\text{pred}}$ that has the lowest Mean Squared Error (MSE) with the ground truth. The loss is then the MSE of this single "winner" scenario:

$$\mathcal{L}_{\text{recon}} = \|\mathbf{y}_{\text{gt}} - \mathbf{y}_{n^*}\|_2^2, n^* = \arg\min_{n=1\ldots N} \|\mathbf{y}_{\text{gt}} - \mathbf{y}_n\|_2^2 \tag{9}$$

*For Probability:* The probability loss, $\mathcal{L}_{\text{prob}}$, trains the probability layer to assign the highest probability to this winner. It is the Cross-Entropy loss between logits vector $\boldsymbol{\pi}$ and winner index $n^*$:

$$\mathcal{L}_{\text{prob}} = \text{CrossEntropy}(\boldsymbol{\pi}, n^*) = -\log\left(\frac{\exp(\pi_{n^*})}{\sum_{j=1}^{N} \exp(\pi_j)}\right) \tag{10}$$

In our experiments, we employ a relaxed variant of the WTA loss (Rupprecht et al., 2017) to further stabilize training. The complete formulation of this loss, including its specific implementation for the multivariate case, is provided in the Appendix C.2.

**Inference.** During inference, the model performs a single forward pass to generate the set of $N$ scenarios, $\mathcal{Y}_{\text{pred}}$, and the logits vector, $\boldsymbol{\pi}$. The logits are then converted into a valid probability vector, $\mathbf{p}$, using the Softmax function as in equation 11. The final output of TimePrism is the complete set of Probabilistic Scenarios, $\{(\mathbf{y}_n, p_n)\}_{n=1}^{N}$.

$$\mathbf{p} = \text{Softmax}(\boldsymbol{\pi}), \quad \text{where } p_n = \frac{\exp(\pi_n)}{\sum_{j=1}^{N} \exp(\pi_j)} \tag{11}$$

## 5 EXPERIMENTS

### 5.1 BASIC SETUP

**Data.** Following the recent benchmark for probabilistic time series forecasting, ProTS (Zhang et al., 2024), we evaluate our model on five datasets: Electricity (Elec.), Exchange (Exch.), Solar(Sol.), Traffic(Traf.), and Wikipedia (Wiki.). These are benchmark datasets taken from the GluonTS library (Alexandrov et al., 2020), preprocessed exactly as in prior works (Gasthaus et al., 2019). A detailed analysis of dataset properties cited from previous work is provided in the Appendix C.1. Following prior work, we set the forecast horizon to 24 (hours) for the hourly datasets (Electricity, Solar, Traffic) and 30 (days) for the daily datasets (Exchange, Wikipedia) (Zhang et al., 2024). For TimePrism, the input length is set equal to the forecast horizon. Other baselines may use longer context lengths as lagged features, for which we adhere to their configurations in prior work (Cortés et al., 2025). Notably, TimePrism achieves strong performance even with less information input. For a comprehensive comparison, we also provide results in the Appendix D.2, where TimePrism uses an input length comparable to that of the baselines.

**Metrics.** As established in our framework, the primary metrics are **Weighted CRPS** and **Distortion**. To measure the *Inference Cost*, we report inference Floating Point Operations (FLOPs). While MSE and Mean Absolute Error (MAE) are not primary indicators for probabilistic forecasting, we include their definitions and normalized results in the Appendix B.3 and D.1 for a comprehensive comparison.

**Baselines.** For a comprehensive comparison, we select seven models covering all three categories discussed in our related work. ETS (Hyndman et al., 2008) serves as a non-neural baseline. DeepAR (Salinas et al., 2020) represents parametric distribution models. TimeGrad (Rasul et al., 2021) is a diffusion-based generative model. TempFlow (Rasul et al., 2020), Transformer TempFlow (Trf.Flow), and TACTiS-2 (Ashok et al., 2023) are structured probabilistic models. Tempflow is implemented with Long Short-Term Memory (LSTM) backbone (Hochreiter & Schmidhuber, 1997) and Trf.Flow is implemented with a Transformer backbone (Vaswani et al., 2017), as in Rasul et al. (2020). Finally, TimeMCL (Cortés et al., 2025) represents multi-choice learning models.

**Training Details.** All models are trained using the Adam optimizer with an initial learning rate of $10^{-3}$ for 200 epochs. Given its lack of hidden layers, the number of scenarios $N$ is the primary tunable hyperparameter for TimePrism. In this section, TimePrism uses $N = 625$ scenarios, composed of $M = 25$ trend and $K = 25$ seasonal components. In practice, if the number of distinct future scenarios is known a priori, $N$ can be set to match this number; otherwise, as in our benchmark datasets, $N$ should be set to a value large enough to allow the model to learn on its own. Further training details and analysis on $N$ are included in the Appendix C.3. To ensure a fair comparison, considering that TimeMCL employs 16 hypotheses in its original implementation, we specifically include a variant with $N = 16$, denoted as **TimePrism-16**. This serves to validate the effectiveness of the new paradigm, demonstrating that it functions effectively with a simple structure and without requiring a large number of parameters.

### 5.2 MAIN RESULTS

Table 1: **CRPS for Probability Absence**. Results on 5 benchmark datasets. We report the mean $\pm$ standard deviation over 3 random seeds. The best and second results are in **bold** and underlined.

| Model | Elec. | Exch. | Sol. | Traf. | Wiki. |
|---|---|---|---|---|---|
| ETS | $0.376 \pm 0.00$ | $1.22 \pm 0.02$ | $0.375 \pm 0.00$ | $0.813 \pm 0.00$ | $4.88 \pm 0.01$ |
| DeepAR | $0.997 \pm 0.03$ | $0.701 \pm 0.00$ | $0.583 \pm 0.02$ | $0.826 \pm 0.01$ | $1.75 \pm 0.30$ |
| TimeGrad | $0.232 \pm 0.00$ | $0.845 \pm 0.24$ | $0.241 \pm 0.00$ | $0.162 \pm 0.00$ | $0.517 \pm 0.02$ |
| TempFlow | $0.316 \pm 0.00$ | $0.669 \pm 0.01$ | $0.272 \pm 0.00$ | $0.601 \pm 0.01$ | $1.26 \pm 0.06$ |
| Trf.Flow | $0.396 \pm 0.08$ | $1.07 \pm 0.17$ | $0.280 \pm 0.02$ | $0.607 \pm 0.01$ | $1.71 \pm 0.12$ |
| TACTiS-2 | $0.299 \pm 0.01$ | $0.648 \pm 0.03$ | $0.236 \pm 0.03$ | $0.257 \pm 0.01$ | **$0.484 \pm 0.00$** |
| TimeMCL | $0.370 \pm 0.01$ | $1.12 \pm 0.15$ | $0.290 \pm 0.03$ | $0.262 \pm 0.01$ | $0.640 \pm 0.03$ |
| TimePrism-16 | $0.414 \pm 0.12$ | $0.611 \pm 0.06$ | $0.137 \pm 0.00$ | $0.159 \pm 0.02$ | $0.654 \pm 0.01$ |
| TimePrism | **$0.133 \pm 0.02$** | **$0.468 \pm 0.01$** | **$0.0852 \pm 0.00$** | **$0.111 \pm 0.00$** | $0.506 \pm 0.00$ |

**Probability Absence and Weighted CRPS.** The limitation of *Probability Absence* means that decision-makers cannot directly assess the likelihood of specific outcomes from sampling-based models. To quantitatively measure the benefit of providing explicit probabilities and evaluate the overall quality of the forecast distribution, we use Weighted CRPS. Table 1 presents the results on five datasets, reported as the mean and standard deviation over three random seeds (3141, 3142, 3143), following prior work. TimePrism achieves state-of-the-art performance on four of the five datasets and secures the second-best result on Wikipedia. Furthermore, a detailed discussion on the applicability of TimePrism is provided in the Appendix C.1.2.

Table 2: **Distortion for Coverage Inadequacy**. Results on 5 benchmark datasets. We report the mean $\pm$ standard deviation over 3 random seeds. The best and second results are in **bold** and underlined.

| Model | Elec. | Exch. | Sol. | Traf. | Wiki. |
|---|---|---|---|---|---|
| ETS | $1.24 \pm 0.02$ | $1.92 \pm 0.06$ | $1.03 \pm 0.00$ | $2.69 \pm 0.01$ | $142 \pm 0.71$ |
| DeepAR | $2.82 \pm 0.11$ | $1.87 \pm 0.03$ | $1.09 \pm 0.02$ | $1.86 \pm 0.09$ | $5.36 \pm 0.42$ |
| TimeGrad | $0.731 \pm 0.02$ | $1.37 \pm 0.17$ | $0.550 \pm 0.03$ | $0.561 \pm 0.02$ | $1.64 \pm 0.03$ |
| TempFlow | $1.41 \pm 0.04$ | $1.32 \pm 0.02$ | $0.515 \pm 0.03$ | $0.981 \pm 0.00$ | $37.8 \pm 6.12$ |
| Trf.Flow | $1.70 \pm 0.28$ | $1.70 \pm 0.22$ | $0.552 \pm 0.04$ | $1.02 \pm 0.01$ | $63.7 \pm 8.02$ |
| TACTiS-2 | $0.674 \pm 0.04$ | $\underline{0.873 \pm 0.04}$ | $0.586 \pm 0.02$ | $0.592 \pm 0.05$ | $1.26 \pm 0.10$ |
| TimeMCL | $\underline{0.607 \pm 0.01}$ | $1.08 \pm 0.08$ | $0.462 \pm 0.04$ | $0.454 \pm 0.00$ | $1.49 \pm 0.30$ |
| TimePrism-16 | $0.911 \pm 0.27$ | $0.920 \pm 0.04$ | $\underline{0.307 \pm 0.04}$ | $\underline{0.346 \pm 0.09}$ | $\underline{1.16 \pm 0.15}$ |
| TimePrism | $\mathbf{0.211 \pm 0.04}$ | $\mathbf{0.595 \pm 0.01}$ | $\mathbf{0.101 \pm 0.03}$ | $\mathbf{0.144 \pm 0.00}$ | $\mathbf{1.04 \pm 0.03}$ |

**Coverage Inadequacy and Distortion.** To assess *Coverage Inadequacy*, we use the Distortion metric, with results presented in Table 2. TimePrism achieves the state-of-the-art result across all five datasets, demonstrating its superior ability to generate a diverse set of scenarios that covers the ground truth. This is because our reconstruction loss, $\mathcal{L}_{\text{recon}}$, allows the model not to be heavily penalized for predicting a plausible but non-realized future, in datasets containing similar histories but diverse futures.

Table 3: **Inference FLOPs.** FLOPs required to generate $S$ forecast samples on the Exchange dataset with batch size $= 1$. The cost for TimeMCL and TimePrism is constant as they produce all scenarios in a single forward pass.

| Sampling $S$ | DeepAR | TimeGrad | TempFlow | Trf.Flow | TACTiS-2 | TimeMCL | TimePrism |
|---|---|---|---|---|---|---|---|
| 1 | $2.9 \times 10^4$ | $1.9 \times 10^8$ | $5.8 \times 10^6$ | $1.4 \times 10^7$ | $2.5 \times 10^7$ | | |
| 10 | $2.9 \times 10^5$ | $1.9 \times 10^9$ | $5.8 \times 10^7$ | $1.3 \times 10^8$ | $1.2 \times 10^8$ | $8.8 \times 10^6$ | $5.1 \times 10^5$ |
| 100 | $2.9 \times 10^6$ | $1.9 \times 10^{10}$ | $5.8 \times 10^8$ | $1.3 \times 10^9$ | $1.1 \times 10^9$ | | |

**Inference Cost and FLOPs.** To evaluate the *Inference Cost*, we compare the FLOPs required by each model to generate a set of $S$ samples, with results shown in Table 3. As demonstrated, the inference cost of TimePrism is constant regardless of the number of samples required, as it generates all $N$ scenarios and their probabilities in a single forward pass. In contrast, the cost for sampling-based models scales with $S$, forcing a direct trade-off between forecast quality and computational efficiency. TimeMCL also generates its full set of hypotheses in a single pass. However, lacking explicit probabilities, its original implementation for CRPS evaluation relies on resampling from this fixed set. For a fair comparison, we also only report the single-pass FLOPs of TimeMCL.

**Overall Comparison.** The CRPS and Distortion results in Tables 1 and 2 are based on $S = 100$ samples for all baselines. At this sampling level, TimePrism is more efficient by one to five orders of magnitude than its most competitive counterparts (TimeGrad, TACTiS-2, and TimeMCL). The results confirm that TimePrism is more efficient than sampling-based models, especially when a large number of samples is needed, highlighting the efficiency of the Probabilistic Scenarios paradigm. This analysis details the trade-off between inference cost and forecast quality, and how our paradigm transcends it.

Table 4: **Impact of Scenario Count ($N$) on Performance and Complexity.** This table presents a systematic ablation study across Electricity, Exchange, and Solar datasets, illustrating how model complexity and forecasting error (CRPS, Distortion) scale with the number of scenarios $N$.

| N | FLOPs | Solar | | Electricity | | Exchange | |
|---|---|---|---|---|---|---|---|
| | | CRPS | Distortion | CRPS | Distortion | CRPS | Distortion |
| 1 | 1.0x | $0.199 \pm 0.00$ | $0.266 \pm 0.00$ | $0.409 \pm 0.01$ | $0.733 \pm 0.03$ | $0.596 \pm 0.00$ | $0.803 \pm 0.00$ |
| 16 | 4.2x | $0.137 \pm 0.00$ | $0.307 \pm 0.03$ | $0.414 \pm 0.10$ | $0.911 \pm 0.22$ | $0.611 \pm 0.04$ | $0.920 \pm 0.04$ |
| 256 | 19.9x | $0.0927 \pm 0.00$ | $0.158 \pm 0.01$ | $0.162 \pm 0.01$ | $0.267 \pm 0.03$ | $0.486 \pm 0.00$ | $0.666 \pm 0.02$ |
| **625** | 34.8x | $0.0852 \pm 0.00$ | $0.101 \pm 0.03$ | $\mathbf{0.133 \pm 0.01}$ | $\mathbf{0.211 \pm 0.03}$ | $0.468 \pm 0.01$ | $0.595 \pm 0.01$ |
| 1024 | 48.3x | $\mathbf{0.0822 \pm 0.00}$ | $\mathbf{0.0917 \pm 0.01}$ | $0.139 \pm 0.01$ | $0.212 \pm 0.02$ | $\mathbf{0.452 \pm 0.00}$ | $\mathbf{0.583 \pm 0.02}$ |

## 5.3 Impact of Scenario Set Size

We conduct a systematic analysis to investigate the trade-off between the scenario set size $N$ and model performance. Table 4 summarizes the results across three representative datasets (Electricity, Exchange, and Solar) with $N$ ranging from 1 to 1024.

**Complexity Scaling.** A key advantage of our combinatorial architecture ($N = M \times K$) is its efficiency. The parameter complexity of the shared basis layers (Trend and Season) scales as $\mathcal{O}(\sqrt{N})$, while only the probability head scales linearly as $\mathcal{O}(N)$. Consequently, the overall model complexity grows favorably between $\mathcal{O}(N^{1/2})$ and $\mathcal{O}(N)$, allowing for large scenario sets.

**Performance Trends.** Increasing $N$ generally leads to lower CRPS and Distortion errors, as a larger discrete set can approximate the continuous probability space with higher fidelity. However, we observe **diminishing returns**: the performance gains tend to plateau around $N = 625$. Beyond this point, the marginal benefit of adding scenarios decreases while the computational cost continues to rise. Based on this equilibrium, we adopted $N = 625$ as the unified setting for our main experiments.

**Dataset Dependence.** The results also indicate that the "saturation point" varies slightly by dataset. For instance, the Solar dataset benefits more from a larger $N$ compared to the Exchange dataset. This suggests that the optimal $N$ is determined by the intrinsic complexity of the data, highlighting the potential for future work on adaptive mechanisms that dynamically adjust $N$.

## 5.4 Visualization and Qualitative Analysis

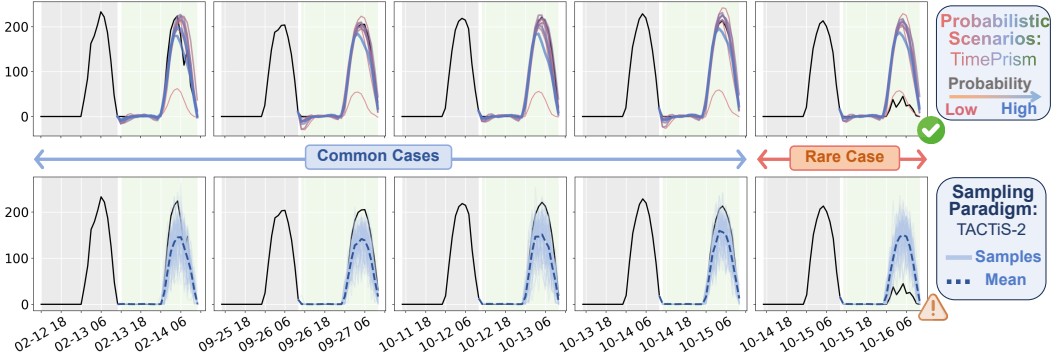

Figure 4: **Qualitative Analysis of the New Paradigm.** A visual comparison between the Probabilistic Scenarios paradigm (TimePrism) and the Sampling Paradigm (TACTiS-2). The figure highlights their distinct behaviors in both common high-peak cases and a rare low-peak case, on the Solar dataset.

To visually compare the two paradigms, we conduct a qualitative analysis on the Solar energy dataset, selecting the last variate ($D = 137$) and identifying instances with similar histories but diverse futures. As shown in Figure 4, these instances include four *Common Cases* of high-peak futures

and one *Rare Case* of a low-peak future. We compare TimePrism against TACTiS-2, the strongest baseline. The top panel displays the top 10 scenarios from TimePrism, with line color and thickness representing probability from low (red, thin) to high (blue, thick). TimePrism successfully captures both types of cases, assigning high probabilities to the common cases while also identifying a rare case with low probability. In contrast, the bottom panel shows that the $S = 100$ samples from TACTiS-2 cluster around their mean (dashed line). While the envelope of TACTiS-2 samples may loosely cover both high-peak and low-peak, its forecast suffers from **Probability Absence**. Without explicit probabilities, the common high peak case can not be distinguished from the rare low peak case, rendering the entire set of samples uninformative for assessment.

## 6 CONCLUSIONS AND DISCUSSION

### 6.1 DISCUSSION

**Reason of Effectiveness.** The strong performance of TimePrism stems from the paradigm's reframing of the learning objective. Instead of learning to model an entire continuous probability space, the model is learning a more structured problem: a probability distribution over a discrete set of scenarios. This concept parallels Vector Quantization (VQ) techniques in representation learning, most notably VQ-VAE (van den Oord et al., 2017), but applies the discretization directly to the output trajectory space rather than a latent space (see Appendix A.3 for a detailed discussion). This shift reduces the required model capacity, allowing a simple linear architecture to achieve strong results.

**Limitations:**

- **Dataset Applicability**. The intentionally simple structure of TimePrism, while effective for validating the paradigm, may have limitations in more complex scenarios, such as those with extremely high dimensionality, or series lacking trend or seasonal patterns.
- **Structural Rigidity.** As a linear model, the current version of TimePrism requires fixed-length inputs and prediction horizons, limiting its flexibility in scenarios where variable-length contexts are available during inference.
- **Simplified Multivariate Modeling.** Our current implementation utilizes a weight-sharing strategy. We believe there is significant room for improvement by incorporating more sophisticated channel-mixing mechanisms to model cross-variate relationships.

**Future Works:**

- **Models within the new Paradigm.** TimePrism serves only as a proof-of-concept. The true potential of Probabilistic Scenarios lies in its application to more powerful backbones. Future work could integrate this paradigm with state-of-the-art architectures like Transformers, Diffusion, or Flow Matching models to unlock new levels of multivariate performance.
- **Refinements of the new Paradigm.** The paradigm itself can be further enhanced. For instance, developing methods to adaptively determine the number of scenarios based on data complexity could improve its practical utility.
- **Decision-Centric Assessment.** Metrics like CRPS and Distortion may not fully reflect the downstream utility of probabilistic forecasts in real-world environments. In future work, decision-centric metrics can be incorporated, such as tail-risk assessment and utility-based scores. Furthermore, we plan to explore the direct integration of our Probabilistic Scenarios paradigm into real-world decision-making to demonstrate its practical value beyond pure forecasting accuracy.

### 6.2 CONCLUSION

Probabilistic time series forecasting is crucial for reliable decision-making. While powerful, current SOTA methods predominantly rely on sampling, a paradigm that faces limitations of *Probability Absence*, *Coverage Inadequacy*, and *Inference Cost*. To address these challenges, we introduced the **Probabilistic Scenarios** paradigm. This paradigm operates by directly producing a set of {Scenario, Probability} pairs in a single forward pass, without reliance on sampling. We validated this new paradigm with TimePrism, a simple linear model. Evaluated under our unified framework, TimePrism addresses these challenges and demonstrates the potential of the new paradigm. In summary, our work provides a practical alternative to sampling and broadens the conceptual landscape of probabilistic forecasting, establishing a promising foundation for future research.

## ETHIC STATEMENT

This paper presents work whose goal is to advance the field of Machine Learning. There are many potential societal consequences of our work, none of which we feel must be specifically highlighted here.

We comply with intellectual property agreements for all data sources. Data are properly anonymized with no concerns regarding sensitive or illegal activity in our dataset.

## REPRODUCIBILITY STATEMENT

The code of this work is available at: `https://github.com/Fifthky/TimePrism`.

## LLM STATEMENT

We utilized a large language model to assist in polishing the grammar and phrasing of our manuscript.

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

# Appendix

## A    THEORETICAL ANALYSIS

### A.1    EFFECTIVENESS OF THE PROBABILISTIC SCENARIOS PARADIGM

The empirical success of the Probabilistic Scenarios paradigm is rooted in its fundamental reframing of the learning objective. This section provides a theoretical perspective on why this reframing leads to a more tractable and effective learning problem.

The conventional sampling-based paradigm requires a model to learn a complex, high-dimensional conditional probability distribution, $P_\theta(\mathbf{y}|\mathbf{x})$, over the continuous space $\mathbb{R}^{T \times D}$. Optimizing this objective, often by maximizing the log-likelihood $\log P_\theta(\mathbf{y}_{\text{gt}}|\mathbf{x})$, is difficult. It requires the model to correctly assign a probability density to every possible point in an infinite space, a task that demands immense model capacity.

In contrast, our Probabilistic Scenarios paradigm transforms this intractable density estimation problem into a more structured, two-part learning task:

1. **Scenario Representation:** The first task is to learn a finite set of $N$ discrete points, $\mathcal{Y}_{\text{pred}} = \{\mathbf{y}_n\}_{n=1}^N$, that effectively represent the most meaningful regions of the true conditional distribution. This simplifies the problem from modeling the entire continuous space to finding a good discrete basis for it.
2. **Probability Assignment:** The second task is to learn a categorical distribution, $\mathbf{p}$, over this finite set of $N$ scenarios. The objective shifts from computing a density $P_\theta(\mathbf{y}_{\text{gt}}|\mathbf{x})$ to solving a large-scale classification problem: determining which of the $N$ representative regions the ground truth $\mathbf{y}_{\text{gt}}$ is most likely to fall into.

In essence, the paradigm decouples the problem of "what" can happen (the scenarios) from "how likely" it is to happen (the probabilities). This structured decomposition significantly reduces the complexity of the learning problem, allowing even simple models to allocate their limited capacity efficiently and achieve strong performance.

### A.2    THEORETICAL FOUNDATIONS OF TIMEPRISM

The theoretical analysis of the Winner-Takes-All principle in this section is inspired by the framework presented in Cortés et al. (2025) and Letzelter et al. (2024). However, we adapt and extend this analysis to our specific non-autoregressive, combinatorial architecture and our probabilistic objective, which, as we will show, provides stronger theoretical guarantees.

#### A.2.1    OPTIMAL SCENARIOS VIA RECONSTRUCTION LOSS

The goal of our reconstruction loss is to find a set of scenarios that provides the best discrete approximation of the continuous space of all possible future trajectories. We formalize this in the following proposition.

**Proposition 1.** *Assuming that the model parameters reach a local minimum of the reconstruction loss, a necessary condition is that the set of $N = M \cdot K$ combined scenarios forms a Centroidal Voronoi Tessellation (CVT) of the space of future trajectories, conditioned on the input history. Specifically, each combined scenario $\mathbf{y}_{t,m} + \mathbf{y}_{s,k}$ converges to the conditional mean of its corresponding Voronoi region.*

*Proof.* The objective is to find the model parameters (which in turn define the scenarios) that minimize the expected reconstruction loss over the data distribution $P(\mathbf{x}, \mathbf{y}_{\text{gt}})$. Our model is non-autoregressive, so the generated scenarios $\{\mathbf{y}_n(\mathbf{x})\}$ are a direct function of the input history $\mathbf{x}$. The expected loss is:

$$\mathbb{E}[\mathcal{L}_{\text{recon}}] = \mathbb{E}_{\mathbf{x}}\left[\mathbb{E}_{\mathbf{y}_{\text{gt}}|\mathbf{x}}\left[\min_{n=1...N}\|\mathbf{y}_{\text{gt}} - \mathbf{y}_n(\mathbf{x})\|_2^2\right]\right] \tag{12}$$

The min operator partitions the space of future trajectories, for a given $\mathbf{x}$, into $N$ Voronoi regions, $\{R_n(\mathbf{x})\}_{n=1}^N$. We formally define the Voronoi region for the $n$-th scenario as $R_n(\mathbf{x}) = \{\mathbf{y} \in \mathbb{R}^{T \times D} \mid \|\mathbf{y} - \mathbf{y}_n(\mathbf{x})\|_2 \le \|\mathbf{y} - \mathbf{y}_j(\mathbf{x})\|_2, \forall j\}$. Each region $R_n(\mathbf{x})$ contains all trajectories $\mathbf{y}_{\text{gt}}$ for which the

$n$-th scenario is the winner. The inner expectation can then be rewritten as a sum of integrals over these regions:

$$\mathbb{E}_{\mathbf{y}_{gt}|\mathbf{x}}\left[\min_{n=1\ldots N}\|\mathbf{y}_{gt}-\mathbf{y}_n(\mathbf{x})\|_2^2\right] = \sum_{n=1}^{N}\int_{R_n(\mathbf{x})}\|\mathbf{y}_{gt}-\mathbf{y}_n(\mathbf{x})\|_2^2\, p(\mathbf{y}_{gt}|\mathbf{x})d\mathbf{y}_{gt} \tag{13}$$

To find the optimal scenarios $\{\mathbf{y}_n(\mathbf{x})\}$, we take the functional derivative of the expected loss with respect to each $\mathbf{y}_n(\mathbf{x})$ and set it to zero. The derivative only affects one term in the summation. Following the derivation in Cortés et al. (2025), the minimum is achieved when:

$$\mathbf{y}_n(\mathbf{x}) = \frac{\int_{R_n(\mathbf{x})}\mathbf{y}_{gt}p(\mathbf{y}_{gt}|\mathbf{x})d\mathbf{y}_{gt}}{\int_{R_n(\mathbf{x})}p(\mathbf{y}_{gt}|\mathbf{x})d\mathbf{y}_{gt}} = \mathbb{E}[\mathbf{y}_{gt}\mid\mathbf{y}_{gt}\in R_n(\mathbf{x})] \tag{14}$$

This derivation holds provided that the Voronoi region has non-zero probability mass, that is, when $\int_{R_n(\mathbf{x})}p(\mathbf{y}_{gt}|\mathbf{x})d\mathbf{y}_{gt} \neq 0$. This demonstrates that for any given history $\mathbf{x}$, the scenarios generated by an optimal model must be the conditional means of their respective **Voronoi regions** (Du et al., 1999). In geometric terms, the set of $N$ scenarios acts as a set of centers that partition the high-dimensional space of all possible futures into $N$ distinct regions, known as a Voronoi tessellation. Each region consists of all future trajectories that are closer to one specific scenario than to any other. Our result shows that the WTA training objective effectively drives the model to find an optimal set of "cluster centers" (our scenarios) that best represent the underlying structure of the data, where "best" is defined in the sense of minimizing the expected squared error, akin to the objective in k-means clustering (Cortés et al., 2025; Arthur & Vassilvitskii, 2007).

### A.2.2 Scenario Representation and Distortion

Our reconstruction loss is designed to optimize for scenario fidelity, which directly contributes to the model's ability to achieve a low Distortion score. The core mechanism lies in how the Winner-Takes-All (WTA) objective interacts with datasets exhibiting diverse potential futures from similar histories.

Consider the gradient of the reconstruction loss, $\mathcal{L}_{recon}$, with respect to the model's parameters $\theta$. The parameters $\theta$ define the mapping from the input $\mathbf{x}$ to the entire set of scenarios $\mathcal{Y}_{pred}(\mathbf{x};\theta) = \{\mathbf{y}_n(\mathbf{x};\theta)\}_{n=1}^N$. The loss for a single data instance $(\mathbf{x},\mathbf{y}_{gt})$ is:

$$\mathcal{L}_{recon}(\theta) = \|\mathbf{y}_{gt}-\mathbf{y}_{n^*}(\mathbf{x};\theta)\|_2^2 \tag{15}$$

where the winner index $n^*$ is itself a function of $\theta$:

$$n^*(\theta) = \arg\min_{n=1\ldots N}\|\mathbf{y}_{gt}-\mathbf{y}_n(\mathbf{x};\theta)\|_2^2 \tag{16}$$

Assuming the winner index $n^*$ is locally constant with respect to small changes in $\theta$, the gradient of the loss is given by the chain rule:

$$\nabla_\theta\mathcal{L}_{recon} = \frac{\partial\mathcal{L}_{recon}}{\partial\mathbf{y}_{n^*}}\cdot\frac{\partial\mathbf{y}_{n^*}(\mathbf{x};\theta)}{\partial\theta} \tag{17}$$

Crucially, for all non-winning scenarios where $n\neq n^*$, the partial derivative of the loss with respect to their outputs is zero:

$$\frac{\partial\mathcal{L}_{recon}}{\partial\mathbf{y}_n} = \mathbf{0}\quad\forall n\neq n^* \tag{18}$$

This implies that the gradients for the parameters governing the non-winning scenarios are also zero for this specific training instance.

The direct consequence of Eq. equation 18 is that the model is not explicitly penalized for generating a plausible but non-realized scenario. In a dataset containing instances of "similar histories, diverse futures," this property allows different scenarios within the set $\mathcal{Y}_{pred}$ to specialize in representing different potential outcomes without interfering with one another during training. For one training instance, only the parameters responsible for the winning scenario are updated to better match the ground truth. For another instance with a similar history but a different future, a different scenario may become the winner, and its corresponding parameters will be updated. This dynamic encourages the model to maintain a diverse and comprehensive set of scenarios to cover the full spectrum of possibilities observed in the training data, directly leading to a lower expected Distortion.

### A.2.3 Optimal Probabilities via Probability Loss

The goal of our probability loss is to ensure that the learned probability vector $\mathbf{p}$ accurately reflects the true probability mass over the Voronoi regions defined by the optimal scenarios.

**Proposition 2.** *At the global minimum of the expected probability loss, the predicted probability vector $\mathbf{p}$ matches the true conditional probability mass function over the Voronoi regions. That is, $p_n = P(\mathbf{y}_{gt} \in R_n \mid \mathbf{x})$, where $R_n$ is the Voronoi region of the $n$-th scenario.*

*Proof.* The optimization objective for the probability loss is to minimize the expected Cross-Entropy loss. We denote the cross-entropy between two discrete distributions $\mathbf{q}$ and $\mathbf{p}$ as $H(\mathbf{q}, \mathbf{p}) = -\sum_n q_n \log p_n$.

$$\mathbb{E}[\mathcal{L}_{\text{prob}}] = \mathbb{E}_{\mathbf{x}} \left[ \mathbb{E}_{\mathbf{y}_{gt}|\mathbf{x}}[-\log p_{n^*(\mathbf{x}, \mathbf{y}_{gt})}(\mathbf{x})] \right] = \mathbb{E}_{\mathbf{x}}[H(\mathbf{q}(\mathbf{x}), \mathbf{p}(\mathbf{x}))] \tag{19}$$

Let $q(n \mid \mathbf{x}) = P(\mathbf{y}_{gt} \in R_n \mid \mathbf{x})$ be the true, unknown probability that the $n$-th scenario is the winner for a given history $\mathbf{x}$. The inner expectation corresponds to the cross-entropy between this true distribution $\mathbf{q}(\mathbf{x})$ and the model's predicted distribution $\mathbf{p}(\mathbf{x}) = \text{Softmax}(\boldsymbol{\pi}(\mathbf{x}))$. By the properties of cross-entropy:

$$\mathbb{E}_{\mathbf{x}}[H(\mathbf{q}(\mathbf{x}), \mathbf{p}(\mathbf{x}))] = \mathbb{E}_{\mathbf{x}}[D_{KL}(\mathbf{q}(\mathbf{x})\|\mathbf{p}(\mathbf{x}))] + \mathbb{E}_{\mathbf{x}}[H(\mathbf{q}(\mathbf{x}))] \tag{20}$$

Since the entropy of the true distribution $H(\mathbf{q}(\mathbf{x}))$ is a constant with respect to our model's parameters, minimizing the expected cross-entropy is equivalent to minimizing the expected KL divergence. The KL divergence is non-negative and is minimized at zero if and only if $\mathbf{p}(\mathbf{x}) = \mathbf{q}(\mathbf{x})$ for all $\mathbf{x}$. Thus, the optimal solution for our probability output is the true probability distribution over the discrete set of winner outcomes.

### A.2.4 Probability Matching and CRPS

Our paradigm's ability to achieve strong performance on the Weighted CRPS metric is rooted in its direct optimization of a true probability distribution. As established in Proposition 2, the Cross-Entropy loss drives the model's output probability vector, $\mathbf{p} = \text{Softmax}(\boldsymbol{\pi})$, to match the true conditional probability mass function over the set of optimal scenarios. The objective is to minimize the Kullback-Leibler (KL) divergence between the predicted and true discrete distributions, $D_{KL}(\mathbf{q}(\mathbf{x})\|\mathbf{p}(\mathbf{x}))$, where $\mathbf{q}(\mathbf{x})$ is the true distribution of winner outcomes. Since the Weighted CRPS directly incorporates the probability vector $\mathbf{p}$ (equation 4), a model that learns a more accurate probability distribution is expected to achieve a lower (better) score.

This approach provides a strong theoretical foundation for probabilistic modeling. The probability $p_n$ for a scenario $\mathbf{y}_n$ in our framework represents a holistic assessment of the entire trajectory, conditioned on the initial history. In contrast, autoregressive multi-hypothesis models like TimeMCL (Cortés et al., 2025), where scenarios, termed hypotheses in the original work, are generated step-by-step, face a challenge in aggregating pointwise confidences into a valid trajectory-level probability. For instance, consider two scenarios over a horizon of $T = 2$. Scenario A might have pointwise confidences of $(0.2, 0.2)$, while Scenario B has $(0.1, 0.3)$. Averaging these values, as is done for evaluation in TimeMCL, would assign both scenarios an identical score of $0.2$. However, under the principles of conditional probability, their joint probabilities would be different $(0.04$ vs. $0.03)$, a distinction that simple averaging fails to capture. Furthermore, the set of scores produced by TimeMCL does not constitute a valid probability distribution as their sum is not constrained to be one.

### A.3 Connection to Discrete Representation Learning

As noted in our discussion on the model's effectiveness, the *Probabilistic Scenarios* paradigm shares conceptual roots with discrete representation learning techniques, most notably Vector Quantized Variational AutoEncoders (VQ-VAE) (van den Oord et al., 2017). Both approaches posit that continuous spaces can be effectively approximated by a finite set of discrete vectors. However, TimePrism distinguishes itself from VQ-VAE in three fundamental aspects, tailored for the forecasting task:

- **Discretization Target:** VQ-VAE discretizes latent features, which serve as intermediate representations. In contrast, TimePrism directly discretizes future trajectories, operating within the final output space.

- **Nature of Codebook/Scenarios:** VQ-VAE utilizes a static, global codebook shared across all inputs, where codes are fixed parameters learned from the entire dataset. Conversely, TimePrism generates a dynamic set of scenarios in real-time based on the input. These scenarios function as a conditional codebook that adapts to the specific history of each time series.

- **Probability Modeling:** VQ-VAE typically employs an implicit, two-stage approach that requires training a separate prior model over discrete codes to perform sampling and probability estimation. TimePrism, however, uses an explicit, end-to-end approach featuring a built-in probability head that directly outputs the probability distribution $p$ over the generated scenarios in a single forward pass.

## B METRICS

### B.1 IMPLEMENTATION DETAILS

This section provides detailed formulations for our primary metrics, Weighted CRPS and Distortion, clarifying how they are applied to the outputs of both the Probabilistic Scenarios and sampling-based paradigms.

**Weighted CRPS.** Our implementation of the Continuous Ranked Probability Score is computed on a per-channel basis. For each variate $d \in \{1, \ldots, D\}$, we calculate the score using the energy score formulation. Given the normalized ground truth for a single channel, $\mathbf{y}'_{\text{gt},d} \in \mathbb{R}^T$, a set of $N$ normalized scenarios for that channel, $\{\mathbf{y}'_{n,d}\}_{n=1}^N$, and a corresponding probability vector for that channel, $\mathbf{p}_d = (p_{1,d}, \ldots, p_{N,d})$, the per-channel Weighted CRPS is:

$$\text{CRPS}_d = \sum_{n=1}^N p_{n,d} \|\mathbf{y}'_{n,d} - \mathbf{y}'_{\text{gt},d}\|_1 - \frac{1}{2} \sum_{n=1}^N \sum_{j=1}^N p_{n,d} p_{j,d} \|\mathbf{y}'_{n,d} - \mathbf{y}'_{j,d}\|_1 \tag{21}$$

where $\| \cdot \|_1$ denotes the L1 norm. The final reported CRPS score is the average of these per-channel scores. For sampling-based models, each of the $S$ samples is assigned a uniform probability $p_{i,d} = 1/S$ for all channels.

**Distortion.** In contrast to CRPS, our Distortion metric is computed jointly across all dimensions to assess the quality of the entire multivariate trajectory. This aligns with its purpose of evaluating the coverage of the joint distribution. It is defined as the minimum Root Mean Squared Error (RMSE) over the set of complete multivariate scenarios:

$$\text{Distortion}(\mathcal{Y}, \mathbf{y}_{\text{gt}}) = \min_{\mathbf{y}_n \in \mathcal{Y}} \sqrt{\frac{1}{T \cdot D} \|\mathbf{y}_n - \mathbf{y}_{\text{gt}}\|_F^2} \tag{22}$$

where $\mathcal{Y}$ represents the set of scenarios and $\| \cdot \|_F$ is the Frobenius norm. Note that the calculation is performed on normalized data as described above. For Probabilistic Scenarios, the minimization is performed over the complete set of $N$ scenarios, $\mathcal{Y} = \mathcal{Y}_{\text{pred}}$. For sampling-based models, it is performed over the set of $S$ generated samples, $\mathcal{Y} = \mathcal{Y}_{\text{samples}}$.

### B.2 COMPREHENSIVENESS AND FAIRNESS

**Scenarios and Probabilities.** Our evaluation framework is comprehensive because its two primary metrics are complementary, addressing the two core components of a probabilistic scenario. The Weighted CRPS evaluates the quality of the entire predictive distribution, considering both the accuracy of the *scenarios* and the correctness of their assigned *probabilities*. Distortion, on the other hand, isolates the quality of the scenario set itself by focusing solely on its best-case coverage, irrespective of probability assignments.

**Per-channel and Joint Evaluation.** While our per-channel CRPS formulation is a standard approach (Zhang et al., 2024), it is known to be insensitive to errors in the correlation structure of a multivariate forecast (Marcotte et al., 2023). We specifically complement this with a jointly computed Distortion metric. Because Distortion evaluates the error over the entire $T \times D$ space for each scenario, it is

sensitive to the quality of the multivariate structure, thus compensating for the limitations of the per-channel CRPS.

**L1 and L2 Norms.** The use of different norms for our two primary metrics is a deliberate design choice. For Weighted CRPS, we use the L1 norm, which is standard for this metric and provides robustness against outliers (Zhang et al., 2024). This is appropriate for a metric assessing the overall distributional quality, where the influence of single extreme errors should be contained. For Distortion, whose sole purpose is to measure the fidelity of the best available scenario, we use the L2 norm (via RMSE), aligned with related work (Cortés et al., 2025). Its higher sensitivity to large deviations is a feature, as it more strictly penalizes a model whose best-case scenario is still far from the ground truth.

**Fairness.** Our evaluation framework is designed to be fair. The Continuous Ranked Probability Score is a strictly proper scoring rule, meaning it is minimized in expectation if and only if the predicted distribution coincides with the true data-generating distribution (Zhang et al., 2024). Our Weighted CRPS, as an average of these strictly proper rules applied to the marginal distributions, inherits this property for the set of marginals. Distortion, however, is not a strictly proper scoring rule as it only considers the single best scenario. For this reason, it serves as a complementary, auxiliary metric focused specifically on coverage, not as a complete measure of probabilistic quality.

### B.3 COMPLEMENTARY METRICS

For a more comprehensive comparison, we also report on two complementary metrics: Mean Squared Error (MSE) and Mean Absolute Error (MAE). These metrics are computed on the same per-channel normalized data as our primary metrics to ensure a consistent evaluation scale. While these are typically used for deterministic forecasting, we include their definitions and results in the Appendix to align with standard practices in recent benchmarks (Zhang et al., 2024; Cortés et al., 2025).

For our Probabilistic Scenarios paradigm, we derive a single representative forecast from the set of scenarios by weighting them by their learned probabilities. For sampling-based models, this is the standard mean or median of the samples.

**Mean Squared Error (MSE).** Following standard practice, the MSE is calculated based on the **mean forecast**, $\hat{\mathbf{y}}_{\text{mean}}$. For a set of scenarios $\mathcal{Y}_{\text{pred}}$ with probabilities $\mathbf{p}$, this is the expectation of the predictive distribution:

$$\hat{\mathbf{y}}_{\text{mean}} = \sum_{n=1}^{N} p_n \mathbf{y}_n \tag{23}$$

The MSE score is then the average of the per-channel Mean Squared Errors:

$$\text{MSE} = \frac{1}{D} \sum_{d=1}^{D} \left( \frac{1}{T} \left\| \mathbf{y}_{\text{gt},d} - \hat{\mathbf{y}}_{\text{mean},d} \right\|_2^2 \right) \tag{24}$$

**Mean Absolute Error (MAE).** The MAE is calculated based on the **median forecast**, $\hat{\mathbf{y}}_{\text{median}}$, which is the 0.5-quantile of the predictive distribution. For a set of scenarios $\mathcal{Y}_{\text{pred}}$ with probabilities $\mathbf{p}$, the weighted median is computed for each point in the trajectory. The MAE score is then the average of the per-channel Mean Absolute Errors, where $\|\cdot\|_1$ denotes the L1 norm:

$$\text{MAE} = \frac{1}{D} \sum_{d=1}^{D} \left( \frac{1}{T} \left\| \mathbf{y}_{\text{gt},d} - \hat{\mathbf{y}}_{\text{median},d} \right\|_1 \right) \tag{25}$$

## C DATA AND EXPERIMENT DETAILS

### C.1 DATA ANALYSIS

#### C.1.1 DATASET PROPERTIES

We evaluate our approach on five widely-used benchmark datasets sourced from the GluonTS library (Alexandrov et al., 2020), with preprocessing consistent with recent work (Cortés et al., 2025). As

Table 5: Dataset characteristics and properties.

| Dataset | Dim. $D$ | Domain $\mathcal{X}$ | Freq. | Time Steps | $T$ | Trend | Seasonality | Non-Gaussianity |
|---|---|---|---|---|---|---|---|---|
| Sol. | 137 | $\mathbb{R}^+$ | Hour | 7,009 | 24 | 0.1688 | 0.8592 | 0.5004 |
| Elec. | 370 | $\mathbb{R}^+$ | Hour | 5,833 | 24 | 0.6443 | 0.8323 | 0.3579 |
| Exch. | 8 | $\mathbb{R}^+$ | Day | 6,071 | 30 | 0.9982 | 0.1256 | 0.2967 |
| Traf. | 963 | $(0, 1)$ | Hour | 4,001 | 24 | 0.2880 | 0.6656 | 0.2991 |
| Wiki. | 2,000 | $\mathbb{N}$ | Day | 792 | 30 | 0.5253 | 0.2234 | 0.2751 |

summarized in Table 5, these datasets span multiple domains and exhibit diverse characteristics in terms of dimensionality (Dim. $D$), data domain ($\mathcal{X}$), frequency, and length. To further characterize the data within the forecast horizon ($T$), we include three quantitative indicators from a recent benchmark, ProbTS (Zhang et al., 2024): trend strength ($F_T$), seasonality strength ($F_S$), and Non-Gaussianity. This selection allows for a comprehensive evaluation across a spectrum of time series properties, from low to high dimensionality and from strong periodicity to trend-dominated behavior.

- **Electricity (Elec.)** contains the hourly power consumption of 370 clients. It exhibits strong seasonality ($F_S = 0.83$) due to daily and weekly human activity patterns, along with a noticeable trend ($F_T = 0.64$).
- **Exchange (Exch.)** records the daily exchange rates of eight currencies. As is common with financial data, it is heavily dominated by trend ($F_T = 0.99$) and shows very weak seasonality ($F_S = 0.13$).
- **Solar (Sol.)** consists of the hourly solar power output from 137 locations. It has the strongest seasonality ($F_S = 0.86$) in our benchmark due to the clear day-night cycle, but a very weak underlying trend ($F_T = 0.17$). It also displays the highest non-Gaussianity.
- **Traffic (Traf.)** measures the hourly occupancy rates of 963 road sensors. It shows moderate seasonality ($F_S = 0.67$) driven by daily rush-hour patterns, coupled with a relatively weak trend ($F_T = 0.29$).
- **Wikipedia (Wiki.)** contains the daily page views for 2000 Wikipedia articles. As the most high-dimensional dataset, its series are characterized by a moderate trend ($F_T = 0.53$) but weak seasonality ($F_S = 0.22$).

### C.1.2 APPLICABILITY OF THE PROPOSED TIMEPRISM

Our proof-of-concept model, TimePrism, is built upon a backbone that decomposes the time series into trend and seasonal components. As shown in Table 5, all five benchmark datasets exhibit a significant presence of either trend or seasonality, providing a solid foundation for this decomposition-based architecture to perform well.

However, it is crucial to distinguish the contributions of the paradigm from those of the specific backbone. The remarkable performance of TimePrism, achieving 9 out of 10 state-of-the-art results, is primarily attributable to the fundamental shift in the learning objective introduced by the Probabilistic Scenarios paradigm. By transforming the complex task of continuous density estimation into a more structured problem of learning a discrete distribution over a combinatorial scenario space, the paradigm itself simplifies the learning challenge. The decomposition backbone merely provides a simple yet effective way to generate the initial candidate scenarios for this paradigm.

Consequently, while the current implementation of TimePrism might be less suitable for datasets where both trend and seasonality are weak, this does not diminish the validity of the underlying paradigm. The Probabilistic Scenarios framework itself makes no assumptions about the data's characteristics and can be integrated with more advanced backbones better suited for different data characteristics in future work.

### C.2 IMPLEMENTATION DETAILS OF PROPOSED TIMEPRISM

This section provides the exact formulations for the loss functions used to train TimePrism in the multivariate setting. The total loss, $\mathcal{L}_{\text{Prism}}$, is the sum of a reconstruction loss and a probability loss.

For the multivariate case, the loss is computed on a per-channel basis and then averaged across all $D$ channels.

For each channel $d \in \{1, \ldots, D\}$, we first identify the channel-specific winner index, $n_d^*$:

$$n_d^* = \arg \min_{n=1\ldots N} \|\mathbf{y}_{\text{gt},d} - \mathbf{y}_{n,d}\|_2^2 \tag{26}$$

The total reconstruction loss, incorporating the Relaxed-WTA mechanism, is the average of the per-channel relaxed losses:

$$\mathcal{L}_{\text{recon}} = \frac{1}{D} \sum_{d=1}^{D} \left[ (1 - \epsilon) \cdot \mathcal{L}_{n_d^*,d} + \frac{\epsilon}{N - 1} \sum_{n \neq n_d^*} \mathcal{L}_{n,d} \right] \tag{27}$$

where $\mathcal{L}_{n,d} = \|\mathbf{y}_{\text{gt},d} - \mathbf{y}_{n,d}\|_2^2$ is the MSE for the $n$-th scenario on the $d$-th channel.

Similarly, the total probability loss is the average of the per-channel Cross-Entropy losses, where each channel's probability distribution is optimized against its own winner:

$$\mathcal{L}_{\text{prob}} = \frac{1}{D} \sum_{d=1}^{D} \text{CrossEntropy}(\boldsymbol{\pi}_d, n_d^*) \tag{28}$$

The following subsections provide a detailed motivation for the two key components of these loss functions: the Relaxed-WTA mechanism and the per-channel, weight-sharing design.

### C.2.1 RELAXED WINNER-TAKES-ALL LOSS

The motivation for the relaxed variant in Eq. equation 27 addresses a potential issue in the standard WTA objective. In the standard formulation ($\epsilon = 0$), non-winning scenarios receive zero gradient for a given training instance. This can lead to parameter stagnation if certain scenarios are consistently not selected as winners across the dataset. By providing a small, non-zero gradient to all non-winning scenarios (controlled by the hyperparameter $\epsilon = 0.01$ in our work), the relaxed loss ensures that all parameters in the scenario-generating layers receive continuous updates, promoting more robust and stable optimization (Rupprecht et al., 2017).

### C.2.2 WEIGHT SHARING

To maintain the structural simplicity and lightweight nature of TimePrism, we adopt a weight-sharing strategy for handling multivariate time series. Instead of learning a separate set of parameters for each of the $D$ variates, the three linear layers in our model (Trend, Season, and Probability layers) share their weights across all variates. This design significantly reduces the total parameter count (Zeng et al., 2023).

As detailed in Eq. equation 28, TimePrism learns a separate probability distribution (parameterized by $\boldsymbol{\pi}_d$) over the shared set of scenarios for each channel, rather than explicitly modeling the joint probability distribution. However, the use of weight sharing allows the model to *implicitly* learn cross-channel relationships during training. Because the weights of the linear layers are shared, the gradient used to update them is an aggregation of the gradients from all $D$ channels. This forces the model to learn a basis of trend and seasonal components, along with their probabilistic mappings, that is collectively useful for the entire multivariate system. Thus, while the model is fully decoupled across channels during inference, the training process is coupled, enabling the simple architecture to capture implicit cross-channel structures. This design choice directly explains the model's performance on the high-dimensional (2000 variates) Wikipedia dataset. The weight-sharing assumption is less likely to hold in datasets with high channel heterogeneity, where each series may follow a distinct pattern. The observed lower performance on this specific dataset is therefore an expected consequence of our intentionally simple, weight-sharing design, rather than a flaw in the Probabilistic Scenarios paradigm itself.

### C.3 TRAINING PROCEDURE

**Baseline Configurations.** The configurations for all baseline models, including DeepAR, TimeGrad, TempFlow, Transformer TempFlow, TACTiS-2, and TimeMCL, adhere to the experimental setups

Table 6: **MAE.** Results on five benchmark datasets, reported as the mean $\pm$ standard deviation over three random seeds. Lower is better. The best result is in **bold**, and the second best is underlined.

| Model | Elec. | Exch. | Sol. | Traf. | Wiki. |
|---|---|---|---|---|---|
| ETS | $0.577 \pm 0.00$ | $1.90 \pm 0.05$ | $0.558 \pm 0.00$ | $1.21 \pm 0.00$ | $4.81 \pm 0.13$ |
| DeepAR | $1.12 \pm 0.02$ | $1.09 \pm 0.01$ | $0.921 \pm 0.04$ | $1.36 \pm 0.05$ | $3.83 \pm 0.96$ |
| TimeGrad | $\underline{0.369 \pm 0.00}$ | $1.33 \pm 0.35$ | $\underline{0.383 \pm 0.00}$ | $\underline{0.278 \pm 0.00}$ | $1.03 \pm 0.03$ |
| TempFlow | $0.633 \pm 0.12$ | $1.73 \pm 0.27$ | $0.451 \pm 0.03$ | $1.02 \pm 0.00$ | $2.74 \pm 0.19$ |
| Trf.Flow | $0.633 \pm 0.12$ | $1.73 \pm 0.27$ | $0.451 \pm 0.03$ | $1.02 \pm 0.00$ | $2.74 \pm 0.19$ |
| Tactis2 | $0.467 \pm 0.03$ | $\underline{1.02 \pm 0.03}$ | $0.388 \pm 0.03$ | $0.420 \pm 0.02$ | $\mathbf{0.944 \pm 0.01}$ |
| TimeMCL | $0.519 \pm 0.00$ | $1.38 \pm 0.21$ | $0.431 \pm 0.04$ | $0.438 \pm 0.03$ | $1.23 \pm 0.09$ |
| TimePrism | $\mathbf{0.171 \pm 0.03}$ | $\mathbf{0.666 \pm 0.01}$ | $\mathbf{0.0832 \pm 0.01}$ | $\mathbf{0.144 \pm 0.00}$ | $\underline{0.995 \pm 0.01}$ |

established in prior work (Cortés et al., 2025), encompassing model architecture, hyperparameters, and other training details. In this work, TimeMCL is configured with $N = 16$ scenarios, consistent with its original implementation (Cortés et al., 2025). We deem this a fair comparison because TimeMCL's autoregressive structure is computationally intensive. Even with only 16 scenarios, its inference FLOPs ($8.8 \times 10^6$) are an order of magnitude higher than TimePrism's with 625 scenarios ($5.1 \times 10^5$). The original work presents two variants, relaxed-WTA (r-WTA) and annealed-WTA (a-WTA). Based on their reported results in Table 1 of their work, the r-WTA variant achieved stronger performance (3 first-place and 2 second-place results versus 2 second-place results for a-WTA). Therefore, we use the more competitive r-WTA variant as our baseline. All other configurations for TimeMCL are kept identical to the original work.

**Batch Size and Scaler.** Following the setup in Cortés et al. (2025), all baselines are trained with a batch size of 200, with the exception of TimeGrad, which uses a batch size of 100 on the Wikipedia dataset due to memory constraints. For TimePrism, we use a batch size of 100 for all datasets except Wikipedia, for which a batch size of 50 is used. While TimePrism has very low inference FLOPs, our intentionally simple implementation is not optimized for memory efficiency, necessitating a slightly smaller batch size on high-dimensional datasets. The data scaler configurations for all baseline models are identical to those used in Cortés et al. (2025). For TimePrism, we use the 'mean_std' scaler for the Exchange dataset and the 'mean' scaler for all other datasets.

**Proposed TimePrism Configuration.** The number of scenarios $N$ in TimePrism is automatically factorized into the two closest integers for the number of trend ($M$) and seasonal ($K$) components. In our main experiments, $N$ is set to 625, corresponding to a configuration of $M = 25$ and $K = 25$. Given the hourly (24) and daily (30) frequencies of our datasets, we set the decomposition kernel size to 7. An analysis of the effect of different values of $N$ on performance is provided in a subsequent appendix.

**Historical Context Length.** Nominally, for datasets sourced from GluonTS, the input look-back length is often set equal to the prediction horizon $T$ (Zhang et al., 2024; Cortés et al., 2025; Alexandrov et al., 2020). However, in practice, some models, like TimeMCL, are designed to use a longer history by incorporating lagged features. Modifying these structural designs to only use an input of length $T$ would be complex and potentially unfair. We therefore adhere to their established configurations. In contrast, our implementation of TimePrism requires only a look-back window of length $T$. It is noteworthy that TimePrism achieves strong results even with less historical information, highlighting the potential of the new paradigm. For a comprehensive comparison, we also provide results in a subsequent appendix where TimePrism uses the full available history as input, which we term "Full History".

Table 7: **MSE.** Results on five benchmark datasets, reported as the mean $\pm$ standard deviation over three random seeds. Lower is better. The best result is in **bold**, and the second best is underlined.

| Model | Elec. | Exch. | Sol. | Traf. | Wiki. |
|---|---|---|---|---|---|
| ETS | $0.519 \pm 0.01$ | $3.96 \pm 0.30$ | $0.455 \pm 0.00$ | $2.09 \pm 0.01$ | $550 \pm 47.70$ |
| DeepAR | $1.42 \pm 0.07$ | $1.47 \pm 0.04$ | $1.19 \pm 0.07$ | $1.87 \pm 0.06$ | $11.5 \pm 4.00$ |
| TimeGrad | $\underline{0.278 \pm 0.01}$ | $2.43 \pm 1.28$ | $\underline{0.361 \pm 0.00}$ | $\underline{0.190 \pm 0.00}$ | $1.84 \pm 0.10$ |
| TempFlow | $3.84 \pm 3.21$ | $4.73 \pm 2.14$ | $0.463 \pm 0.07$ | $1.04 \pm 0.01$ | $676 \pm 302.55$ |
| Trf.Flow | $3.84 \pm 3.21$ | $4.73 \pm 2.14$ | $0.463 \pm 0.07$ | $1.04 \pm 0.01$ | $676 \pm 302.55$ |
| Tactis2 | $0.366 \pm 0.03$ | $\underline{1.23 \pm 0.07}$ | $0.365 \pm 0.06$ | $0.368 \pm 0.02$ | $\underline{1.34 \pm 0.19}$ |
| TimeMCL | $0.393 \pm 0.01$ | $2.46 \pm 1.11$ | $0.542 \pm 0.13$ | $0.319 \pm 0.02$ | $13.7 \pm 18.85$ |
| TimePrism | $\mathbf{0.104 \pm 0.02}$ | $\mathbf{0.712 \pm 0.09}$ | $\mathbf{0.0769 \pm 0.01}$ | $\mathbf{0.0983 \pm 0.01}$ | $\mathbf{1.28 \pm 0.02}$ |

# D   ADDITIONAL EXPERIMENTS

## D.1   RESULTS OF COMPLEMENTARY METRICS

**MAE.** Table 6 presents the results for the Mean Absolute Error, reported as the mean $\pm$ standard deviation over three random seeds (3141, 3142, 3143). As both MAE and our primary metric, CRPS, are based on the L1 norm, the overall ranking of the models shows a similar pattern. TimePrism achieves the best performance on four out of five datasets and the second-best on Wikipedia, reinforcing the conclusions from our main results and demonstrating its strong performance in terms of the median forecast.

**MSE.** The Mean Squared Error results are presented in Table 7, reported as the mean $\pm$ standard deviation over three random seeds (3141, 3142, 3143). As a metric based on the L2 norm, the MSE is more sensitive to large errors or outliers. The results show a consistent pattern where TimePrism outperforms all baselines across all five datasets, demonstrating the robustness of its mean forecast even under a stricter, squared-error evaluation.

## D.2   EXPERIMENTS ON HISTORY LENGTH CONFIGURATION

As discussed in the main text, some baseline models, such as TimeMCL (Cortés et al., 2025), are structurally designed to utilize a historical context longer than the nominal forecast horizon $T$ by incorporating lagged features. Modifying these established architectures to only use an input of length $T$ would be complex and potentially unfair. It is noteworthy that the main results for TimePrism are achieved using only this nominal input length $T$, demonstrating the potential of the new paradigm even with less information.

For a more direct comparison, we present an additional experiment in Table 8 where TimePrism uses the full available history, a variant we term "Full History" (Full His.). The length of this history is set to be comparable to the total context available to the baselines' data processing modules as in Cortés et al. (2025). The results show that using a longer history does not consistently improve TimePrism's performance; in some cases, the scores are similar or slightly worse, though still highly competitive. This is not a perfectly fair comparison, as other models are designed with feature engineering capabilities to extract value from long lagged inputs, while our simple linear model uses the full history directly. For such a simple architecture, a much longer input sequence can introduce noise without a sophisticated mechanism to filter it, which explains why more data does not necessarily lead to better performance.

This highlights a potential direction for future work, where more advanced feature engineering or model structures could be integrated within our paradigm to better leverage longer historical contexts.

Table 8: **Main Results on Primary Metrics with Full History.** Comparison of Weighted CRPS and Distortion on five benchmark datasets. Lower is better. The best result is in **bold**, and the second best is underlined. TimePrism (Full His.) refers to our model using the full historical context for a more direct comparison with baselines.

| Model | Elec. | | Exch. | | Sol. | | Traf. | | Wiki. | |
|---|---|---|---|---|---|---|---|---|---|---|
| | CRPS | Dis. | CRPS | Dis. | CRPS | Dis. | CRPS | Dis. | CRPS | Dis. |
| ETS | 0.376 | 1.23 | 1.23 | 1.98 | 0.374 | 1.03 | 0.815 | 2.69 | 4.88 | 142 |
| DeepAR | 0.993 | 2.79 | 0.698 | 1.89 | 0.607 | 1.11 | 0.829 | 1.82 | 1.41 | 4.88 |
| TimeGrad | 0.230 | 0.720 | 0.739 | 1.33 | 0.237 | 0.587 | 0.163 | 0.540 | 0.516 | 1.62 |
| TempFlow | 0.449 | 1.73 | 0.988 | 1.55 | 0.278 | 0.555 | 0.613 | 1.01 | 1.81 | 71.6 |
| Trf.Flow | 0.449 | 1.73 | 0.988 | 1.55 | 0.278 | 0.555 | 0.613 | 1.01 | 1.81 | 71.6 |
| Tactis2 | 0.285 | 0.637 | 0.641 | 0.919 | 0.222 | 0.567 | 0.243 | 0.55 | **0.481** | 1.37 |
| TimeMCL | 0.375 | 0.603 | 1.30 | 1.10 | 0.301 | 0.485 | 0.251 | 0.455 | 0.624 | 1.32 |
| TimePrism | **0.148** | **0.237** | **0.456** | **0.588** | **0.0835** | **0.140** | **0.109** | **0.140** | 0.508 | **1.01** |
| TimePrism (Full His.) | 0.210 | 0.475 | 0.461 | 0.594 | 0.224 | 0.295 | 0.184 | 0.346 | 0.505 | 1.025 |

Table 9: **Generalizability Analysis with Transformer Backbone.** Comparison of Primary Metrics (CRPS and Distortion) across five datasets. **TimePrism-iT** represents the iTransformer (Liu et al., 2023) architecture adapted to our Probabilistic Scenarios paradigm. All experiments use Seed 3141.

| Model | Elec. | | Exch. | | Sol. | | Traf. | | Wiki. | |
|---|---|---|---|---|---|---|---|---|---|---|
| | CRPS | Dis. | CRPS | Dis. | CRPS | Dis. | CRPS | Dis. | CRPS | Dis. |
| DeepAR | 0.993 | 2.79 | 0.698 | 1.89 | 0.607 | 1.11 | 0.829 | 1.82 | 1.41 | 4.88 |
| TimeGrad | 0.230 | 0.720 | 0.739 | 1.33 | 0.237 | 0.587 | 0.163 | 0.540 | 0.516 | 1.62 |
| TempFlow | 0.449 | 1.73 | 0.988 | 1.55 | 0.278 | 0.555 | 0.613 | 1.01 | 1.81 | 71.6 |
| Trf.Flow | 0.449 | 1.73 | 0.988 | 1.55 | 0.278 | 0.555 | 0.613 | 1.01 | 1.81 | 71.6 |
| Tactis2 | 0.285 | 0.637 | 0.641 | 0.919 | 0.222 | 0.567 | 0.243 | 0.55 | **0.481** | 1.37 |
| TimeMCL | 0.375 | 0.603 | 1.30 | 1.10 | 0.301 | 0.485 | 0.251 | 0.455 | 0.624 | 1.32 |
| TimePrism | **0.148** | **0.237** | 0.456 | **0.588** | **0.0835** | **0.140** | **0.109** | **0.140** | 0.508 | **1.01** |
| TimePrism-iT | 0.330 | 0.600 | **0.454** | 0.681 | 0.164 | 0.245 | 0.201 | 0.371 | 0.756 | 1.425 |

### D.3 PARADIGM GENERALIZABILITY: ADAPTING TO TRANSFORMER ARCHITECTURES

To more rigorously validate that the superior performance of our method stems from the proposed **Probabilistic Scenarios** paradigm rather than solely the specific linear architecture of TimePrism, we conducted a controlled study adapting a distinct, complex architecture to our framework. We selected one of the state-of-the-art Transformer-based time series models, iTransformer (Liu et al., 2023), as the backbone.

**Experimental Setup.** We developed a variant named TimePrism-iT, where the linear encoder of TimePrism is replaced by the inverted Transformer structure from Liu et al. (2023). Crucially, to demonstrate the "out-of-the-box" applicability and robustness of our paradigm, we *did not* perform extensive hyperparameter tuning for TimePrism-iT. Instead, we applied a generally consistent configuration across all datasets. This setup serves as a rigorous stress test to verify if the paradigm can yield performance gains without relying on architecture-specific optimization.

**Results Analysis.** The comparative results are presented in Table 9. Despite being an unoptimized implementation, TimePrism-iT demonstrates remarkable performance. It outperforms standard

Table 10: The results of models in datasets from GIFT-Eval (Aksu et al., 2024) and fev-bench Shchur et al. (2025). The best result in each column is in **bold**.

| Dataset | UCI | | Hosp. | | Hier. | | M-Den. | |
|---|---|---|---|---|---|---|---|---|
| Metrics | CRPS | Distortion | CRPS | Distortion | CRPS | Distortion | CRPS | Distortion |
| ETS | 0.450 | 0.811 | 0.585 | 1.30 | 0.990 | 4.03 | 0.782 | 2.23 |
| Tactis2 | 0.605 | 0.787 | 0.583 | 1.20 | 0.623 | 1.58 | **0.614** | 1.13 |
| TimeMCL | 0.359 | 0.449 | 0.722 | 1.13 | 1.08 | 1.33 | 0.771 | **0.597** |
| TimePrism | **0.261** | **0.394** | **0.565** | **1.06** | **0.602** | **1.03** | 0.907 | 1.11 |

baselines on 6 out of 10 metrics across the five datasets. Notably, on the **Exchange** dataset, TimePrism-iT achieves a CRPS of **0.454**, slightly surpassing even the original linear TimePrism (0.456).

### D.4 EXTENDED EVALUATION ON ADDITIONAL BENCHMARKS

To provide a more comprehensive evaluation of our proposed paradigm, we extended our experiments to include four datasets selected from two latest benchmarks: **Gift-Eval** (Aksu et al., 2024) and **fev-bench** (Shchur et al., 2025). These datasets were chosen to cover diverse domains: **Hierarchical Sales** (Retail, abbr. Hier.), **M-DENSE** (Mobility, abbr. M-Den.), **Hospital Admissions** (Healthcare, abbr. Hosp.), and **UCI Air Quality** (Nature, abbr. UCI).

**Experimental Setup.** For these experiments, we selected the numerical baseline **ETS** and the two most competitive neural models from Table 1 and Table 2, namely **TimeMCL** and **Tactis2**, for comparison. All models were evaluated using a random seed of 3141 to ensure reproducibility.

**Results Analysis.** As shown in the additional results in Table 10, TimePrism maintains its strong performance across these new domains. Regarding the M-DENSE dataset, we observed that TimePrism exhibits relatively higher distortion. We hypothesize that the nature of this dataset may be more suitable for RNN backbones, as both Tactis-2 and TimePrism perform suboptimally on this dataset, while TimeMCL remains competitive. This is not a limitation of our new paradigm, but rather a consequence of TimePrism's simple structure. Nevertheless, achieving SOTA results in 15 out of 18 metrics across 9 datasets still demonstrates the effectiveness of the TimePrism model and highlights the potential of the new paradigm.

### D.5 PROBABILITY CALIBRATION DIAGNOSTICS

To rigorously assess the reliability of the probabilities assigned by TimePrism, we employ two standard diagnostic tools: the **Reliability Diagram** (Coverage vs. Nominal Confidence) and the **Probability Integral Transform (PIT) Histogram**.

**Methodology.** Since TimePrism outputs a tuple $(\mathcal{Y}_{\text{pred}}, \mathbf{p})$ consisting of a finite set of scenarios $\mathcal{Y}_{\text{pred}} = \{\mathbf{y}_n\}_{n=1}^N$ and their associated probabilities $\mathbf{p} = (p_1, \ldots, p_N)$, we compute these metrics as follows:

- **PIT Histogram:** For a ground truth observation $\mathbf{y}_{\text{gt}}$, the PIT value is the cumulative probability of scenarios that are less than or equal to the observation: $\text{PIT} = \sum_{n=1}^N p_n \cdot \mathbb{I}(\mathbf{y}_n \leq \mathbf{y}_{\text{gt}})$. For a perfectly calibrated model, the distribution of PIT values over the test set should approach a Uniform distribution $U[0, 1]$, resulting in a flat histogram.

- **Reliability Diagram:** We calculate the empirical coverage for varying nominal confidence levels $\alpha \in [0, 1]$. The prediction interval for a level $\alpha$ is constructed by aggregating the scenarios $\mathbf{y}_n$ with the highest probabilities until their cumulative sum reaches $\alpha$. If the model is well-calibrated, the curve should align with the diagonal $y = x$. Curves above the diagonal indicate under-confidence (conservative), while curves below indicate over-confidence.

**Analysis.** We performed these diagnostics on two representative datasets: Exchange and Solar. The results are visualized in Figure 5.

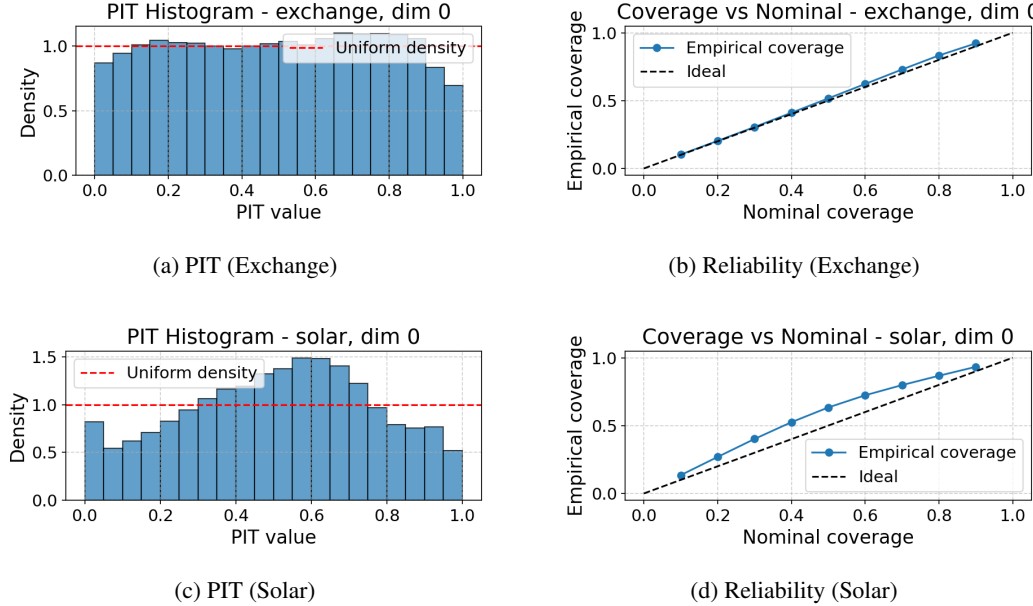

Figure 5: **Calibration Diagnostics.** The diagnostics show different behaviors across datasets: (a)(b) Exchange dataset demonstrates near-perfect calibration; (c)(d) Solar dataset exhibits a slightly conservative profile to ensure robust tail coverage.

- **Exchange Dataset (Fig. 5a & 5b):** The diagnostics indicate near-perfect calibration. The PIT histogram is remarkably flat, and the Reliability Diagram closely follows the ideal diagonal line. This suggests that for stable financial data, TimePrism accurately estimates the true uncertainty distribution.

- **Solar Dataset (Fig. 5c & 5d):** The diagnostics exhibit a slightly conservative profile. The PIT histogram shows a mild hump shape, and the Reliability curve lies slightly above the diagonal. This behavior is expected and often desirable for highly stochastic, multimodal data like Solar energy. It indicates that TimePrism tends to widen its predicted scenario distribution to safely encompass multimodality and potential outliers. This "conservative" strategy ensures robust coverage of low-probability, high-impact tail events without becoming over-confident, aligning with our design goal of prioritizing coverage adequacy.

## D.6 ADDITIONAL VISUALIZATION AND QUALITATIVE ANALYSIS

**Window Selection Rule.** To provide a fair and insightful qualitative comparison, we developed a systematic rule for selecting the windows to be visualized. For a given dataset and variate, we first select a query window from a recent part of the historical data. We then search through the entire history to find the five past windows that are most similar to this query window, based on Euclidean distance. To ensure that the selected windows represent distinct, non-overlapping events, we enforce a minimum temporal separation between them. This greedy, iterative process allows us to identify a set of instances where the model is repeatedly faced with a similar historical context, providing a controlled setting to analyze its predictive behavior.

### D.6.1 VISUALIZATIONS ON OTHER DATASETS

To further demonstrate the applicability of our paradigm, we provide additional qualitative results for TimePrism on the Electricity and Traffic datasets in Figure 6. The top panel showcases forecasts for the Electricity dataset. Across the selected windows, the model successfully generates a diverse set of scenarios that cover the volatile and complex patterns of power consumption, assigning higher probabilities (thicker, blue lines) to the most plausible outcomes. The bottom panel of Figure 6 displays the results for the Traffic dataset. Here, the model also produces a sharp and well-calibrated set of scenarios that effectively captures the distinct peaks and troughs characteristic of traffic flow

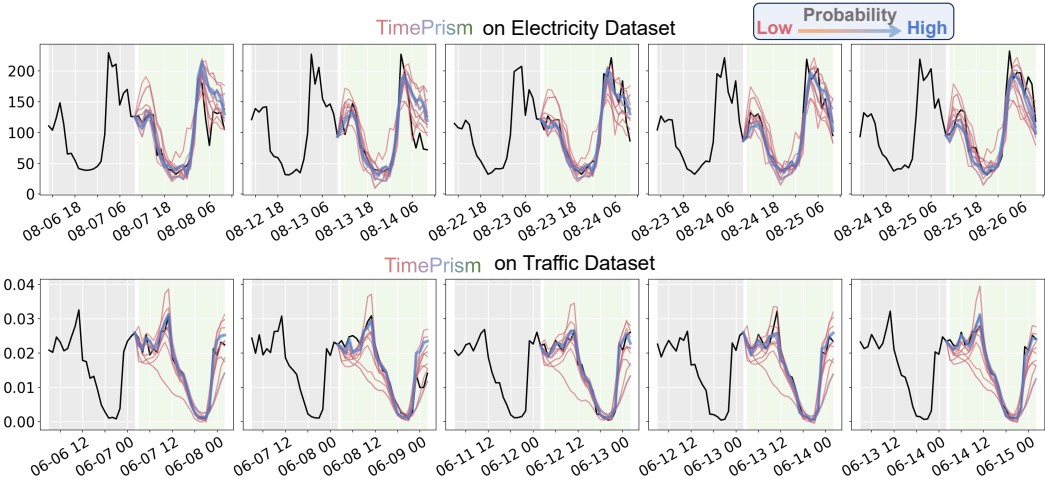

Figure 6: **Qualitative Analysis on Electricity and Traffic Datasets.** Visualization of TimePrism's probabilistic scenarios on two additional benchmark datasets. The top row shows forecasts for the Electricity dataset, and the bottom row shows forecasts for the Traffic dataset.

data. These visualizations further confirm that the Probabilistic Scenarios paradigm can generate meaningful forecasts across different domains and data characteristics.

### D.6.2 FULL COMPARISON ON SOLAR

We now present a full visual comparison of all neural network-based baselines against TimePrism on the Solar dataset. We select two representative variates for this analysis: the first ($D = 1$) and the last ($D = 137$). The figures display the top 10 scenarios from TimePrism, with line color and thickness representing probability from low (red, thin) to high (blue, thick), and 100 samples from each baseline model. The historical context is shown with a gray background, while the future prediction horizon has a light green background.

Figure 7 shows the results for the first variate of the Solar dataset. Across the five selected windows, we observe several *Common Cases* of high-peak solar generation, along with two *Rare Cases* (third and fourth from the left) that exhibit more volatile or lower-peak behavior. For the common cases, TimePrism correctly assigns high probabilities (thicker, blue lines) to scenarios that accurately match the ground truth. Crucially, for the rare cases, it successfully identifies and covers these less frequent patterns while correctly assigning them lower probabilities (thinner, redder lines). In contrast, the sampling-based models, including the strong baseline TACTiS-2, tend to produce a cloud of samples centered around an average forecast. This often results in a mean forecast that matches neither the common nor the rare cases well, and the sample envelope may fail to adequately cover the true outcome in the rare cases, demonstrating the limitations of **Probability Absence** and **Coverage Inadequacy**.

Figure 8 presents the analysis for the last variate of the dataset. This example provides a clear distinction between four *Common Cases* and one *Rare Case* (far right). TimePrism again demonstrates the strength of the Probabilistic Scenarios paradigm: it allocates the majority of its probability mass to accurately predict the common high-peak cases, while still generating a low-probability scenario that correctly captures the rare low-peak future. The sampling-based models, however, struggle with this scenario. Their samples tend to cluster around a mean that represents an uninformative compromise between the high and low peaks. This visually exemplifies how a forecast lacking explicit probabilities can fail to provide actionable insights for decision-making, especially when preparing for rare but critical outcomes.

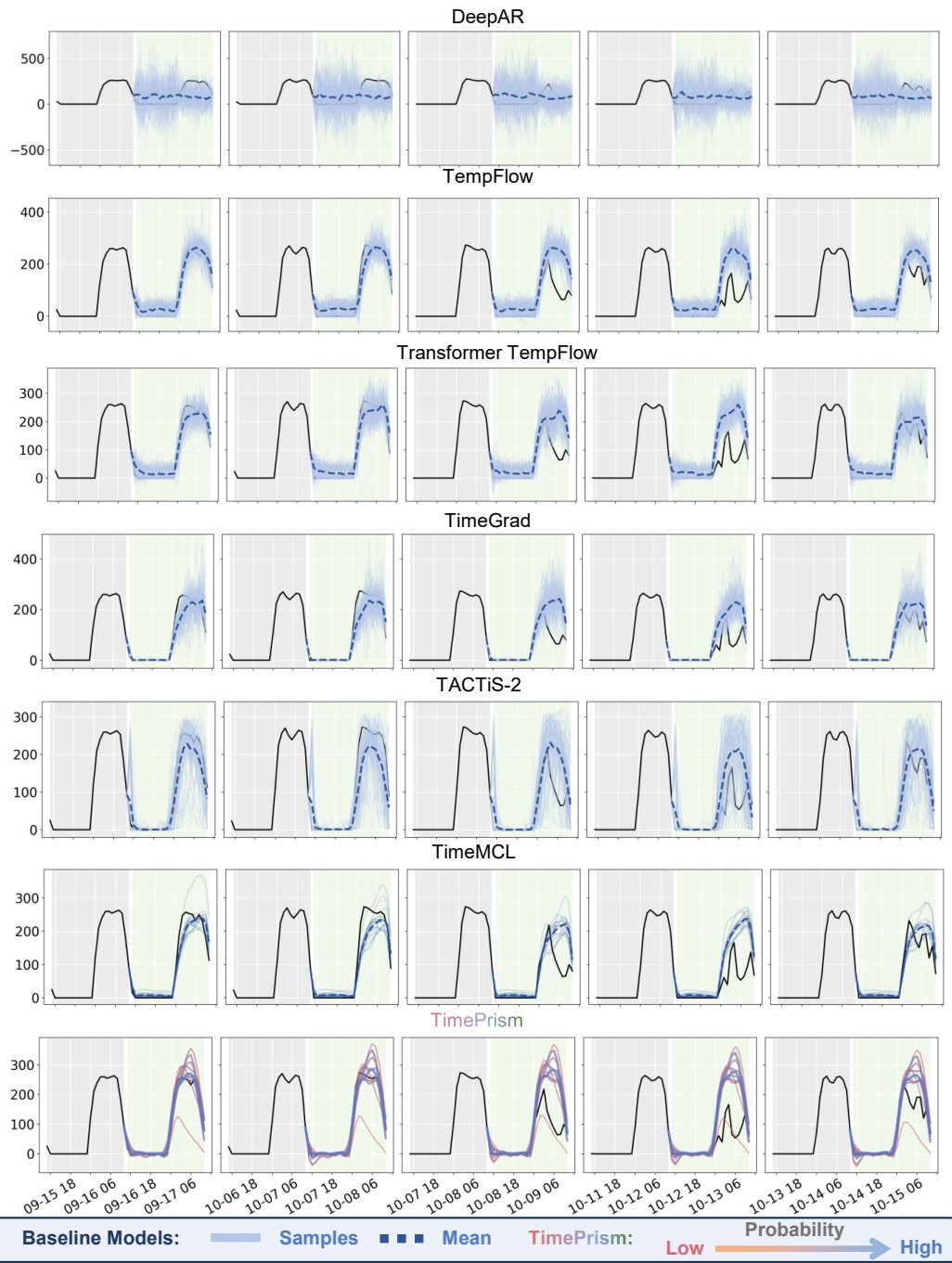

Figure 7: **Qualitative Analysis on Solar (D=1).** A visual comparison of forecasts from all neural network-based models on the first variate of the Solar dataset. The figure highlights performance on both common high-peak cases and two rare cases.

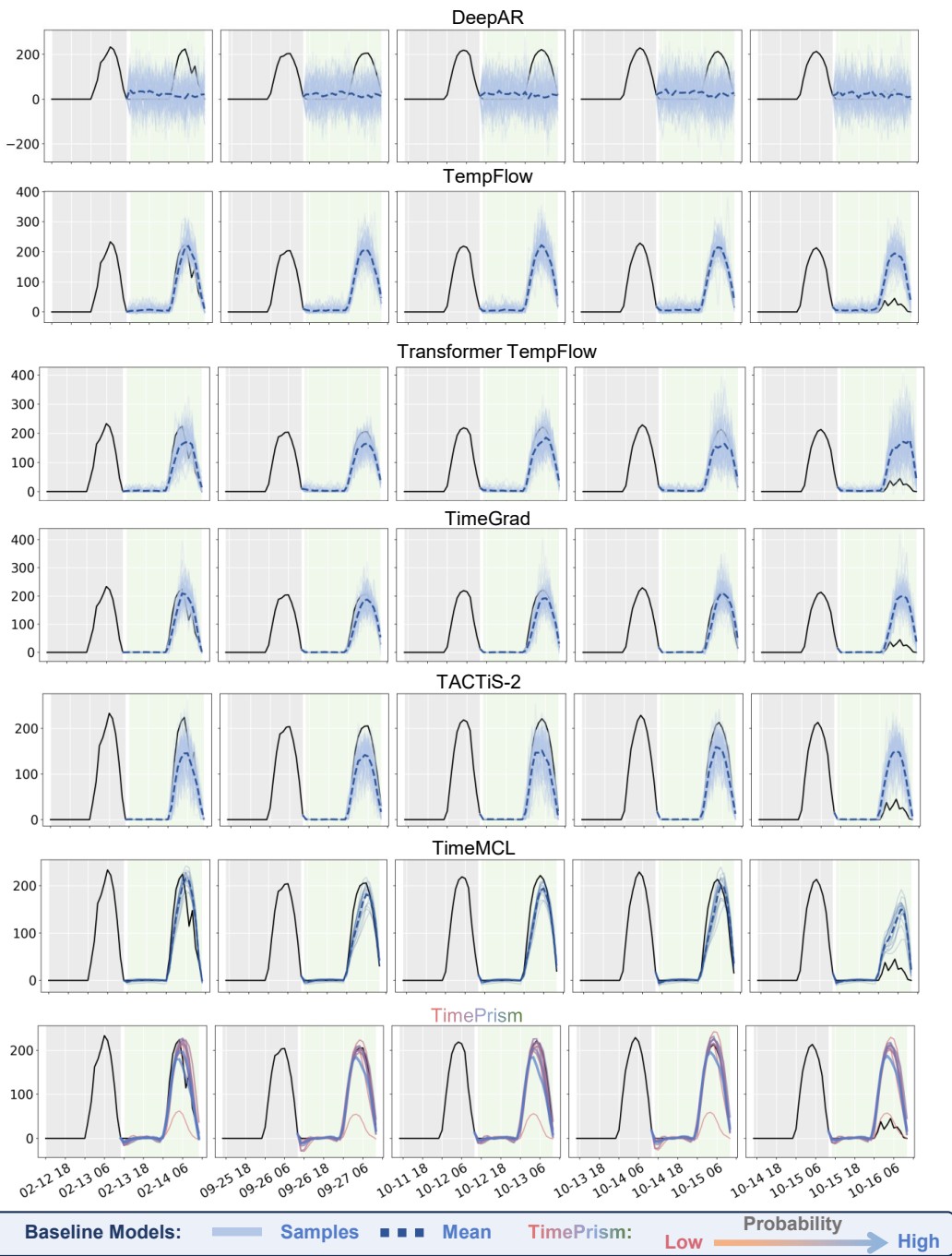

Figure 8: **Qualitative Analysis on Solar (D=137).** A visual comparison on the last variate of the Solar dataset. This case clearly distinguishes between four common high-peak cases and one rare low-peak case.

