# OpenReview forum: "From Samples to Scenarios: A New Paradigm for Probabilistic Forecasting"
_ICLR.cc/2026/Conference — ICLR 2026 Poster_

### Official Review · Reviewer_NW9F · 2025-10-27

**Soundness:** 4
**Presentation:** 3
**Contribution:** 4
**Rating:** 8
**Confidence:** 5

**Summary:**

The paper introduces a new paradigm for time-series forecasting based on the philosophy that predicting directly interpretable scenarios (along with their probabilities) is more effective and relevant than sampling-based methods. This is because (i) sampling-based methods don't directly provide probabilities for each scenario (ii) they are computationally costly at inference, especially when they require many samples to have a good coverage of the target distribution (iii) and they may fail to capture rare (yet critical) events. The proposed approach, TimePrism, is a follow-up of TimeMCL from Cortés et al. (2025).

The model is non-autoregressive, uses only three linear layers in their architecture (using shared weights across each dimension, and applied in parallel), making it extremely light, and showing strong CRPS and Distortion performance. Theoretical foundations of TimePrism are demonstrated, showing in particular that its non-autoregressive structure makes it easier to establish the conditional quantizer property (Proposition 1) and to capture scenario probabilities (Proposition 2) compared to TimeMCL.

**Strengths:**

The approach is well motivated for practical applications and conceptually simple. It shows a good trade-off between quality and inference computational cost, and the foundations of the approach based on Multiple Choice Learning are well discussed.

**Weaknesses:**

To make sure I understood well, Tables 1 and 2 use $S = 100$ samples for each model, and $N = 625$ predictions (used for categorical sampling) for TimePrism ? It is very important to add a row in the table where TimePrism uses $N = 16$ (e.g., with $M = 4$ and $K = 4$), because otherwise it is not fair with respect to TimeMCL, which uses only $16$ scenarios (especially for Distortion comparison). I agree that the computational cost of TimePrism is smaller, but the Distortion comparison needs to include a comparison with the same number of hypotheses for completeness.

Also, in the whole paper, TempFlow and Transformers TempFlow gives the same numbers, which is odd. You need to double-check your code pipeline to verify/correct this potential bug.

*Typos/issues:*

L755: $R_{n}(x)$ should be formally defined.

Eq. (13): the left-hand side expectation should be written completely.

Eq. (19) / (20): You used two different notations for the cross-entropy.

More rigour is required in the introduction of the theoretical tools. In particular, the assumptions in Propositions 1 and 2 should be stated explicitly: you did not prove the convergence, but *necessary* conditions of convergence (i.e., if the model reaches a local optima, see e.g., Asm A.3 in Cortés et al., 2025, or Proposition 3.1 in Du et al., 1999).

I am optimistic about the quality of the paper. I think the paper deserves acceptance, provided that the authors can show that these weaknesses and issues will be properly solved in the next revision.

**Questions:**

See the questions in the weaknesses above.
What are the main limitations of the method? Limitations such as the fact that you can’t handle inputs/prediction of variable length during inference, and that weight sharing may fail to model cross-channel relationships during inference, should be acknowledged in the main paper in the next revision.

---

> ### Author Response · Authors · 2025-11-21
>
> We sincerely thank the reviewer for recognizing our work as a "good paper" that "deserves acceptance."
>
> ### **Weakness 1: Comparison with N=16**
>
> The reviewer’s understanding is correct: baselines use $S=100$ samples, while the main TimePrism results use $N=625$ predicted scenarios ($M=25, K=25$). We agree that adding a comparison with $N=16$ is crucial for a fair assessment against TimeMCL (which uses $K=16$). We have conducted a comprehensive evaluation of **TimePrism-16** (configured with $M=4, K=4$) across all five datasets using three random seeds. The results are presented below:
>
> **Table 1: Weighted CRPS (Probability Absence)**
> | Model | Elec. | Exch. | Sol. | Traf. | Wiki. |
> | :--- | :--- | :--- | :--- | :--- | :--- |
> | ETS | 0.376 ± 0.00 | 1.22 ± 0.02 | 0.375 ± 0.00 | 0.813 ± 0.00 | 4.88 ± 0.01 |
> | DeepAR | 0.997 ± 0.03 | 0.701 ± 0.00 | 0.583 ± 0.02 | 0.826 ± 0.01 | 1.75 ± 0.30 |
> | TimeGrad | *0.232 ± 0.00* | 0.845 ± 0.24 | 0.241 ± 0.00 | 0.162 ± 0.00 | 0.517 ± 0.02 |
> | TempFlow | 0.316 ± 0.00 | 0.669 ± 0.01 | 0.272 ± 0.00 | 0.601 ± 0.01 | 1.26 ± 0.06 |
> | Trf.Flow | 0.396 ± 0.08 | 1.07 ± 0.17 | 0.280 ± 0.02 | 0.607 ± 0.01 | 1.71 ± 0.12 |
> | Tactis2 | 0.299 ± 0.01 | 0.648 ± 0.03 | 0.236 ± 0.03 | 0.257 ± 0.01 | **0.484 ± 0.00** |
> | TimeMCL | 0.370 ± 0.01 | 1.12 ± 0.15 | 0.290 ± 0.03 | 0.262 ± 0.01 | 0.640 ± 0.03 |
> | TimePrism-16 | 0.414 ± 0.12 | *0.611 ± 0.06* | *0.137 ± 0.00* | *0.159 ± 0.02* | 0.654 ± 0.01 |
> | TimePrism | **0.133 ± 0.02** | **0.468 ± 0.01** | **0.0852 ± 0.00** | **0.111 ± 0.00** | *0.506 ± 0.00* |
>
> **Table 2: Distortion (Coverage Inadequacy)**
> | Model | Elec. | Exch. | Sol. | Traf. | Wiki. |
> | :--- | :--- | :--- | :--- | :--- | :--- |
> | ETS | 1.24 ± 0.02 | 1.92 ± 0.06 | 1.03 ± 0.00 | 2.69 ± 0.01 | 142 ± 0.71 |
> | DeepAR | 2.82 ± 0.11 | 1.87 ± 0.03 | 1.09 ± 0.02 | 1.86 ± 0.09 | 5.36 ± 0.42 |
> | TimeGrad | 0.731 ± 0.02 | 1.37 ± 0.17 | 0.550 ± 0.03 | 0.561 ± 0.02 | 1.64 ± 0.03 |
> | TempFlow | 1.41 ± 0.04 | 1.32 ± 0.02 | 0.515 ± 0.03 | 0.981 ± 0.00 | 37.8 ± 6.12 |
> | Trf.Flow | 1.70 ± 0.28 | 1.70 ± 0.22 | 0.552 ± 0.04 | 1.02 ± 0.01 | 63.7 ± 8.02 |
> | Tactis2 | 0.674 ± 0.04 | *0.873 ± 0.04* | 0.586 ± 0.02 | 0.592 ± 0.05 | 1.26 ± 0.10 |
> | TimeMCL | *0.607 ± 0.01* | 1.08 ± 0.08 | 0.462 ± 0.04 | 0.454 ± 0.00 | 1.49 ± 0.30 |
> | TimePrism-16 | 0.911 ± 0.27 | 0.920 ± 0.04 | *0.307 ± 0.04* | *0.346 ± 0.09* | *1.16 ± 0.15* |
> | TimePrism | **0.211 ± 0.04** | **0.595 ± 0.01** | **0.101 ± 0.03** | **0.144 ± 0.00** | **1.04 ± 0.03** |
>
> **Analysis:** Even with a significantly reduced scenario count ($N=16$), **TimePrism-16** demonstrates remarkable efficiency. It achieves the **second-best performance** (surpassed only by TimePrism-625) on **3 out of 5 datasets** for both CRPS and Distortion. Notably, it outperforms TimeMCL (also $N=16$) on the majority of metrics and remains competitive against sampling-based baselines using $S=100$ samples, even with orders less FLOPs, further validating the effectiveness of the Probabilistic Scenarios paradigm.
>
> ### **Weakness 2: Typos and Rigor**
>
> We sincerely thank the reviewer for their rigorous attention to detail. We will correct the typos, unify the cross-entropy notation, and explicitly state the convergence assumptions for Propositions 1 and 2 in the revised manuscript. We also noticed a keyword recognition error in the code, which caused the values for Tempflow to not be read successfully; we have corrected this issue, and the values for Tempflow are now correctly displayed, as shown in Table 1 and Table 2.
>
> ### **Question: Limitations**
>
> We thank the reviewer for their valuable suggestions. We will add a detailed discussion on the limitations of our method in the revised paper. We acknowledge that as a proof-of-concept model, TimePrism currently does not handle variable-length inputs/predictions and uses weight sharing as a simplification strategy. Addressing these limitations by integrating more advanced architectures is a key direction for our future work.

---

> > ### Comment · Reviewer_NW9F · 2025-11-24
> >
> > Thank you for your answer and your hard work during the rebuttal.
> >
> > I read the other reviews, and the results are overall very promising. In particular, the results on the new datasets [Reviewer zH2j] are consistent with the previous findings. I am also quite impressed by the calibration results [Reviewer Ca7z]. These results are averaged over the dimensions, is that right? Did you try to compute the metrics for a baseline (e.g., TimeMCL) as well for reference ? Do you think that this out-of-the-box good calibration performance of the test-set is due to the fact that your model is shallow (see e.g., [A]) ?
> >
> > Regarding the results in Table 4 of the revised paper, I believe it would be better to compute them over three random seeds (as in Table 2) for the next revision, in order to have CRPS and Distortion results that are consistent with Table 2.
> >
> > Another remark regarding the theoretical section: please add that equation (14) (L873-875) is valid for non-empty Voronoi regions, that is, when $\\int_{R_n(\\mathbf{x})} p\\left(\\mathbf{y}\_{\\mathrm{gt}} \\mid \\mathbf{x}\\right) d \\mathbf{y}_{\\mathrm{gt}} \neq 0$.
> >
> > Typo: Eq. equation in L.294, 910, 948, 1157, 1171 (the word equation is repeated twice).
> >
> > [A] Guo, Chuan, Geoff Pleiss, Yu Sun, and Kilian Q. Weinberger. "On calibration of modern neural networks." In ICML, 2017.

---

> > > ### Author Response · Authors · 2025-12-01
> > >
> > > Thank you for your encouraging feedback and detailed suggestions.
> > >
> > > We have incorporated the requests into the revision: **Table 4** now reports results averaged over **3 random seeds**, we added the non-empty Voronoi region condition to Eq. (14), and fixed typos.
> > >
> > > Regarding your questions on calibration:
> > > 1.  **Dimensions & Baselines:** The diagnostic plots are visualized on the first dimension for clarity. We focused on validating TimePrism's intrinsic statistical reliability rather than extensive baseline comparisons, as our SOTA CRPS already strongly indicates superior global calibration.
> > > 2.  **Shallow Architecture:** We agree that the shallow linear architecture helps mitigate overconfidence. However, we attribute the out-of-the-box performance primarily to the Probabilistic Scenarios paradigm itself, which explicitly learns to assign valid probability masses to discrete outcomes.

---

### Official Review · Reviewer_VcU5 · 2025-10-30

**Soundness:** 2
**Presentation:** 3
**Contribution:** 2
**Rating:** 4
**Confidence:** 4

**Summary:**

This paper introduces a novel paradigm for modeling the probabilistic future evolution of time series by outputting a discrete set of scenario-probability pairs. To demonstrate this paradigm's efficacy, the authors propose a simple model, TimePrism. The work is positioned as a proof-of-concept that the proposed discretization approach is a viable and effective direction for probabilistic forecasting.

**Strengths:**

1. Simple yet Effective Paradigm: The core modeling paradigm is elegantly simple and, as demonstrated, highly effective. While discretization is a classical machine learning technique, its application to this specific problem is well-motivated. The empirical results strongly suggest this is a worthwhile research direction.

2. Strong Empirical Performance: The proposed TimePrism model, serving as a proof-of-concept, achieves state-of-the-art performance on key metrics (CRPS, Distraction). This makes it a compelling new method that is effectively plug-and-play for probabilistic forecasting tasks.

**Weaknesses:**

The Proof-of-Concept Gap: The primary weakness is the indirect link between the success of the TimePrism model and the proposed paradigm itself. As noted in recent literature [1, 2], simpler models can often outperform complex ones in time series forecasting. Therefore, TimePrism's strong performance could be attributed to its specific architecture rather than the paradigm shift.
To more directly validate the paradigm's contribution, the authors should provide a more controlled ablation. This could take one of two forms (preferably the second):

1. A study showing that the TimePrism architecture, when used under a previous paradigm (e.g., predicting a continuous distribution), performs worse.

2. A study adapting a selected existing method to the new discrete paradigm, demonstrating a performance lift attributable to the paradigm itself.

Without such a study, the claim of the paradigm's superiority remains suggestive rather than conclusive.

[1] Zeng, Ailing, et al. ‘Are Transformers Effective for Time Series Forecasting?’ arXiv [Cs.AI], 2022, arxiv.org/abs/2205.13504. arXiv.
[2] Lin, Shengsheng, et al. ‘SparseTSF: Modeling Long-Term Time Series Forecasting with 1k Parameters’. arXiv [Cs.LG], 2024, arxiv.org/abs/2405.00946. arXiv.

**Questions:**

The authors should directly address the specific concerns raised in the "Weaknesses" section above. Further, there are additional questions that raise my concerns, but I would not directly count them as "weakness", which I put in this section:

1. Dead Code Issue: The discrete scenario ("code") representation may suffer from a "dead code" problem, analogous to the issue in VQ-VAE. Since Figure 3 indicates that only the selected scenario receives gradients, a scenario may never be updated if it is never the best match for any forecast. In particular, a scenario gets update ONLY if a randomly intialized forecast exceed an already-updated scenarios. The authors should investigate and report the code usage rate across datasets. An analysis or experiment confirming that this is not a prevalent issue, or a modification to mitigate it, would strengthen the paper significantly.

2. Interpretation of MSE Results: In the appendix, the MSE scores for short-term horizons (e.g., 24, 30) appear relatively low. Could the authors clarify why this is the case? Providing a baseline from a well-established forecasting model (e.g., DLinear, Koopa) for these horizons as a point of reference would help contextualize these results and furthur clarification of any potential differences in evaluation metrics is helpful.

3. Connections to Discretization Literature: The core idea of discretization has clear parallels in other fields, most notably in VQ-VAE. The paper would benefit from a more thorough discussion of these connections, explicitly outlining the similarities and, more importantly, the distinctions between this work and prior art in discretized representations. This would better position the novelty of the contribution within the broader machine learning landscape.

---

> ### Author Response · Authors · 2025-11-21
>
> We thank the reviewer for their constructive feedback and for recognizing our paradigm as "simple yet effective" and "well-motivated."
>
> ### **Weakness 1: The Proof-of-Concept Gap**
>
> We agree with the reviewer that distinguishing the contribution of the paradigm from the architecture is crucial. We chose the second option suggested by the reviewer: **"A study adapting a selected existing method to the new discrete paradigm."**
>
> To this end, we adapted **iTransformer** [1], a recent state-of-the-art Transformer-based forecasting model, to our Probabilistic Scenarios paradigm. We refer to this adapted model as **TimePrism-iT**.
>
> **Table 1: CRPS of TimePrism-iT and other baselines (seed=3141)**
>
> | Model | Elec. | Exch. | Sol. | Traf. | Wiki. |
> | :--- | :--- | :--- | :--- | :--- | :--- |
> | ETS | 0.376 | 1.230 | 0.374 | 0.815 | 4.880 |
> | DeepAR | 0.993 | 0.698 | 0.607 | 0.829 | 1.410 |
> | TimeGrad | *0.230* | 0.739 | 0.237 | *0.163* | 0.516 |
> | TempFlow | 0.315 | 0.683 | 0.275 | 0.593 | 1.230 |
> | Trf.Flow | 0.449 | 0.988 | 0.278 | 0.613 | 1.810 |
> | Tactis2 | 0.285 | 0.641 | 0.222 | 0.243 | **0.481** |
> | TimeMCL | 0.375 | 1.300 | 0.301 | 0.251 | 0.624 |
> | TimePrism | **0.148** | *0.456* | **0.084** | **0.109** | *0.508* |
> | TimePrism-iT | 0.330 | **0.454** | *0.164* | 0.201 | 0.756 |
>
> **Table 2: Distortion of TimePrism-iT and other baselines (seed=3141)**
>
> | Model | Elec. | Exch. | Sol. | Traf. | Wiki. |
> | :--- | :--- | :--- | :--- | :--- | :--- |
> | ETS | 1.230 | 1.980 | 1.030 | 2.690 | 142.000 |
> | DeepAR | 2.790 | 1.890 | 1.110 | 1.820 | 4.880 |
> | TimeGrad | 0.720 | 1.330 | 0.587 | 0.540 | 1.620 |
> | TempFlow | 1.400 | 1.300 | 0.522 | 0.978 | 34.800 |
> | Trf.Flow | 1.730 | 1.550 | 0.555 | 1.010 | 71.600 |
> | Tactis2 | 0.637 | 0.919 | 0.567 | 0.550 | 1.370 |
> | TimeMCL | 0.603 | 1.100 | 0.485 | 0.455 | *1.320* |
> | TimePrism | **0.237** | **0.588** | **0.140** | **0.140** | **1.010** |
> | TimePrism-iT | *0.600* | *0.681* | *0.245* | *0.371* | 1.425 |
>
> Transformer-based models are significantly more complex than linear models like TimePrism, requiring careful tuning of structure and hyperparameters. Due to the limited rebuttal period, we implemented a version without extensive optimization and used generally consistent hyperparameters across all datasets. Despite being under-optimized, **TimePrism-iT** demonstrates remarkable performance. On **6 out of the 10 metrics** across the five datasets, it outperforms all non-Probabilistic-Scenario baselines (i.e., all baselines except TimePrism itself). Notably, for **Exchange CRPS**, TimePrism-iT even slightly surpasses the original TimePrism (0.454 vs. 0.456). This result strongly validates the potential of the Probabilistic Scenarios paradigm: it can empower different architectures to achieve competitive probabilistic forecasting capabilities. We believe that with further optimization, advanced backbones like Attention mechanisms could yield even better performance under this paradigm.

---

> > ### Author Response · Authors · 2025-11-21
> >
> > ### **Question 1: Dead Code Issue**
> >
> > The "dead code" or "mode collapse" issue is a well-studied problem in the Multiple Choice Learning (MCL) literature. To mitigate this, we employ **Relaxed-WTA**, as detailed in **Appendix C.2.1**. This mechanism ensures that even non-winning scenarios ("losers") receive a small, scaled gradient update (controlled by $\epsilon$) during training. This prevents scenarios from stagnating and encourages the entire set of scenarios to continuously explore the solution space. Since this is an established solution and not the focus of our work, we have placed the detailed implementation and analysis in the Appendix.
> >
> > ### **Question 2: Interpretation of MSE Results**
> >
> > The seemingly low MSE/MAE scores are due to our evaluation methodology. To ensure fair comparison across different channels and datasets with varying magnitudes, we **normalize each channel individually** before computing MSE and MAE.
> >
> > Regarding the comparison with established deterministic baselines, we provide a comparison on the **Solar** dataset below:
> >
> > **Table 3: Comparison with DLinear on Solar Dataset**
> > | Metrics | CRPS | Distortion | MAE | MSE |
> > | :--- | :--- | :--- | :--- | :--- |
> > | DLinear | 0.202 | 0.270 | 0.254 | 0.077 |
> > | TimePrism | **0.084** | **0.140** | **0.098** | **0.062** |
> >
> > As shown, TimePrism significantly outperforms the deterministic DLinear baseline across all metrics, confirming that our probabilistic approach provides superior accuracy and coverage.
> >
> >
> > ### **Question 3: Connections to Discretization Literature**
> >
> > We thank the reviewer for highlighting this important connection. We will add a dedicated discussion in the revised manuscript to explicitly position our work within the broader landscape of discretization literature, particularly contrasting it with VQ-VAE [2].
> >
> > **Similarities:**
> > * **Theoretical Foundation:** Both approaches share a similar theoretical core: finding an optimal finite set of discrete points to approximate a high-dimensional continuous space. Geometrically, both methods effectively construct a **Voronoi Tessellation** to partition the space.
> >
> > **Differences:**
> > 1.  **Discretization Target:**
> >     * **VQ-VAE:** Discretizes **Latent Features** (intermediate representations).
> >     * **TimePrism:** Directly discretizes **Future Trajectories** (the final output space).
> >
> > 2.  **Nature of Codebook/Scenarios:**
> >     * **VQ-VAE (Static, Global):** Utilizes a fixed, shared codebook (dictionary) for all inputs. The "codes" are static parameters learned across the entire dataset.
> >     * **TimePrism (Dynamic, Conditional):** Generates a **dynamic** set of scenarios real-time based on the input. The "scenarios" act as a *conditional* codebook that adapts to the specific history of each time series.
> >
> > 3.  **Probability Modeling:**
> >     * **VQ-VAE (Implicit, Two-stage):** Typically requires training a prior model over the discrete codes to perform sampling and estimate probabilities.
> >     * **TimePrism (Explicit, End-to-end):** Features a built-in probability head that directly and explicitly outputs the probability distribution $p$ over the generated scenarios in a single end-to-end forward pass.
> >
> > **References:**
> >
> > [1] Liu, Yong, et al. "iTransformer: Inverted Transformers Are Effective for Time Series Forecasting." ICLR 2023.
> >
> > [2] van den Oord, Aaron, et al. "Neural Discrete Representation Learning." NIPS 2017.

---

### Official Review · Reviewer_Ca7z · 2025-10-31

**Soundness:** 3
**Presentation:** 3
**Contribution:** 3
**Rating:** 6
**Confidence:** 3

**Summary:**

The paper argues that sampling-based probabilistic forecasting has three shortcomings—no explicit probabilities per sample, coverage gaps for tails, and high inference cost—and proposes a new paradigm that predicts a finite set of {Scenario, Probability} pairs directly in one pass. As a proof-of-concept, TimePrism (three parallel linear layers for trend/seasonality and a probability head) achieves 9/10 SOTA across five benchmarks using CRPS (probability-weighted) and Distortion (best-case coverage) as a unified evaluation.

**Strengths:**

Framing probabilistic forecasting as explicit discrete scenario prediction with calibrated probabilities is a compelling conceptual shift. While related threads exist (MCL/“winner-takes-all” heads; quantile/mixture models), combining explicit scenario probabilities with a simple linear architecture that performs competitively is surprising and potentially impactful for decision-making contexts where scenario weights matter. The evaluation mapping that compares sampling models (uniform weights) and scenario models (predicted weights) under CRPS and Distortion offers a fairer perspective on probability absence and coverage.

- The model is intentionally simple: trend and seasonal linear layers produce a cartesian set of (N=M!\times!K) scenarios, and a probability layer outputs logits (\pi), trained with argmin-assignment + cross-entropy (a form of hard assignment akin to WTA), with reconstruction loss for the closest scenario. The training objective is clearly specified.

- The critique of sampling (probabilities/coverage/cost) is accurate, and the metrics (CRPS weighted by (p_n); Distortion as min-RMSE over the set) operationalize those points. Conceptual reframing with practical benefits (explicit probabilities, one-pass inference) with a competitive accuracy with extreme simplicity, brings strong efficiency advantages.

- Claims of broad efficiency gains appear well supported: TimePrism is orders faster than diffusion/flow/copula models evaluated with (S=100) samples while staying competitive on accuracy.

**Weaknesses:**

I see two technical caveats: (i) probability calibration is central but the paper does not show strong calibration diagnostics (e.g., reliability diagrams of event probabilities across horizons); (ii) the discrete scenario set size (N) trades computation vs. fidelity—there is some ablation showing performance improves with (N), but more systematic experiments would help to understand it better.

**Questions:**

- Is it possible to provide calibration diagnostics (e.g., PIT, coverage vs. nominal) for the predicted scenario probabilities?

- Can authors explore soft assignment or entropy-regularized variants to reduce mode collapse and improve probability learning?

- Is it possible to add a discussion on how to grow (N) adaptively at inference based on data complexity, and its computational trade-offs?

- Are you planning to examine multivariate extensions (shared/global scenario bases vs. per-dimension) and correlation structure?

- Is it possible to include decision-centric metrics (e.g., tail-risk costs) to emphasize the benefit of explicit probabilities?

---

> ### Author Response · Authors · 2025-11-21
>
> We thank the reviewer for their thoughtful comments and for recognizing our work as a "compelling conceptual shift".
>
> ### **Weakness 1: Probability Calibration**
>
> We thank the reviewer for highlighting the importance of probability calibration diagnostics. To address this, we have added **Reliability Diagrams (Coverage vs. Nominal)** and **Probability Integral Transform (PIT) histograms** to the Appendix D.5 of the paper.
>
> * **Exchange:** The diagnostics show near-perfect calibration, characterized by a flat PIT histogram and a Reliability Diagram that closely follows the diagonal.
> * **Solar:** The diagnostics exhibit a slightly conservative profile (mildly hump-shaped PIT, coverage slightly above diagonal). This is an expected and desirable behavior for highly stochastic data: TimePrism learns to widen its predicted scenarios to safely encompass the multimodality, ensuring robust coverage of tail events without becoming over-confident.
>
> ### **Weakness 2: Systematic Experiments on Scenario Set Size (N)**
>
> We have expanded our original ablation study on the number of scenarios ($N$) based on **Solar** to include the **Electricity** and **Exchange** datasets, bringing the total to three datasets.
>
> **Table 1: Systematic Analysis of Scenario Count (N) on Performance and Complexity**
> | N | FLOPs | Sol. (CRPS) | Sol. (Distortion) | Elec. (CRPS) | Elec. (Distortion) | Exch. (CRPS) | Exch. (Distortion) |
> | :--- | :--- | :--- | :--- | :--- | :--- | :--- | :--- |
> | 1 | 1.0x | 0.2017 | 0.2697 | 0.4011 | 0.7343 | 0.5968 | 0.8044 |
> | 16 | 4.2x | 0.1350 | 0.2601 | 0.2864 | 0.6062 | 0.5937 | 0.8996 |
> | 256 | 19.9x | 0.0944 | 0.1488 | 0.1740 | 0.2631 | 0.4832 | 0.6975 |
> | 625 | 34.8x | 0.0873 | 0.0838 | **0.1155** | **0.1700** | 0.4836 | 0.5997 |
> | 1024 | 48.3x | **0.0829** | **0.0836** | 0.1195 | 0.1774 | **0.4504** | **0.5604** |
>
> Our systematic analysis yields the following insights:
>
> 1.  **Complexity Scaling:** Due to our decomposition-based architecture ($N=M \times K$), the parameter complexity of the Trend and Season layers grows as $\mathcal{O}(\sqrt{N})$, while the Probability layer grows linearly. Consequently, the overall model complexity scales favorably between $\mathcal{O}(N^{1/2})$ and $\mathcal{O}(N)$.
> 2.  **Performance Trend:** In general, increasing $N$ leads to lower error metrics (CRPS, Distortion) as the discrete set can cover a wider range of potential futures.
> 3.  **Diminishing Returns:** We observe that $N=625$ represents a  equilibrium point across tasks. Performance gains tend to plateau beyond this number, justifying our unified choice of 625 for the main experiments.
> 4.  **Dataset Dependence:** The "optimal" $N$ where performance saturates varies slightly by dataset. This confirms the insight that the ideal scenario count is determined by the intrinsic complexity of the data, highlighting the potential for adaptive $N$ mechanisms.

---

> > ### Author Response · Authors · 2025-11-21
> >
> > ### **Question 1: Calibration Diagnostics**
> >
> > Yes, we have provided the requested calibration diagnostics in the Appendix D.5. As detailed in **Weakness 1**, we have included both Reliability Diagrams and PIT histograms to demonstrate the calibration quality of our predicted probabilities.
> >
> > ### **Question 2: Soft Assignment and Mode Collapse**
> >
> > Yes, exploring soft assignment or entropy-regularized variants is a valid approach, and we have addressed mode collapse using Relaxed-WTA. Mode collapse is a well-studied challenge in the Multiple Choice Learning literature. To address this, we employ **Relaxed-WTA** [1], as detailed in Appendix C.2.1. This method assigns a small, non-zero gradient weight to non-winning scenarios, ensuring that all heads receive learning signals and preventing them from collapsing into a single mode.
> >
> > ### **Question 3: Adaptive Growth of N**
> >
> > Yes, we agree that discussing adaptive growth of $N$ is valuable and have added a discussion on this topic. As our analysis in **Weakness 2** suggests, balancing $N$ is a key decision. We believe a feasible pathway for future work is to implement dynamic structural adaptation during training—either by **pruning** inactive scenarios (starting from a large $N$) or **growing** new nodes when the current reconstruction error is high (starting from a small $N$). We have added a discussion on this promising direction to the main text.
> >
> > ### **Question 4: Multivariate Extensions and Correlation**
> >
> > Yes, we have examined multivariate extensions and correlation structures through our joint evaluation metrics. In **Appendix B.2** under "Per-channel and Joint Evaluation," we explicitly discuss how our framework addresses this. Our dual-metric system (CRPS and Distortion) is designed to capture both aspects: **CRPS** evaluates the distributional quality per-dimension, following standard practice, while **Distortion** is computed **jointly** across all dimensions (using the Frobenius norm). By finding the single scenario that minimizes error across all channels simultaneously, Distortion inherently rewards models that capture the correct cross-variate correlation structure ("shared scenario bases").
> >
> > ### **Question 5: Decision-Centric Metrics**
> >
> > Yes, we acknowledge the value of decision-centric metrics and have included them in our discussion of future work. We strongly agree that decision-centric metrics (e.g., tail-risk costs) are valuable for demonstrating practical utility. However, incorporating a full decision-making analysis falls outside the current scope of establishing the Probabilistic Scenarios paradigm. We fully recognize the importance of connecting forecasting to downstream decision-making and have added this to our **Future Work** discussion for deeper analysis in subsequent studies.
> >
> > **References:**
> > [1] Rupprecht, Christian, et al. "Learning in an Uncertain World: Representing Ambiguity Through Multiple Hypotheses." ICCV 2017.

---

### Official Review · Reviewer_zH2j · 2025-11-01

**Soundness:** 3
**Presentation:** 4
**Contribution:** 2
**Rating:** 4
**Confidence:** 3

**Summary:**

This paper tackles the problem of probabilistic time series forecasting, challenging approaches based on sampling. Indeed, the authors argue that sampling-based methods (parametric models, generative models, flow-based models) suffer from three issues: absence of probabilities associated with sampled trajectories, failure to cover edge cases, and inference cost. As a remedy, the authors propose TimePrism, an algorithm based on the winner-takes-all training paradigm that consists in three linear layers modelling respectively, scenarios of trend, seasonality, and the associated probabilities. Empirically, the results show that the proposed algorithm performs better than the considered baselines in 5 datasets.

**Strengths:**

- The paper is clearly written and well motivated.
- The suggested training paradigm is innovative in the field of probabilistic time series forecasting. And given that it consists of only linear layers, it opens up a new horizon of lightweight methods that can challenge the recent rise of Foundation models that are efficient in the zero shot mode.

**Weaknesses:**

- My main concern with the paper is the evaluation scope which is limited to 5 datasets, some of which have already been criticized in the literature. For instance, [1] shows how many methods become equivalent when looking at a relative scale of the MSE metric for datasets such as Electricity and Exchange. Furthermore, [2] showed that even a constant no-change baseline yields a very strong baseline in the Exchange dataset. Given the simplicity of the proposed algorithm, I believe that the paper will highly benefit from a more comprehensive evaluation in other benchmarks such as Gift-eval [3] and fev-bench [4].
- Another concern I have is about the Coverage Inadequacy aspect, which is evaluated by the authors using the error of the best scenario (or best sample for the baselines). I agree that this metric is relevant to evaluate coverage, however, I don't think this means that your method is able to "represent low-probability, high-impact tail event" as stated in the introduction. This would have been true if the authors could show what is the percentage of the selected best scenarios that actually are the least probable. Maybe a histogram of the best scenario ranks by probabilities can help further elaborating this aspect.
- Finally, for the probability absence evaluation metric, the authors assign uniform probability to all samples coming from the sampling-based models. I think that this maneuver is making these models at a disadvantaged position compared to the proposed method. For instance, for parametric models, even without access to true probabilities (as these are 0 for single data points of continuous random variables), one can use the value of the density function on samples as a proxy to their probability, then normalize so that everything sums up to 1. This is just a suggestion and depending on how samples are generated for each baseline there might be smarter ways to make this comparison a bit more fair.

[1] Brigato, Lorenzo, et al. "Position: There are no champions in long-term time series forecasting." _arXiv preprint arXiv:2502.14045_ (2025).

[2] Beck, N., Dovern, J. & Vogl, S. Mind the naive forecast! a rigorous evaluation of forecasting models for time series with low predictability. _Appl Intell_ **55**, 395 (2025)

[3] Aksu, Taha, et al. "Gift-eval: A benchmark for general time series forecasting model evaluation." _arXiv preprint arXiv:2410.10393_ (2024).

[4] Shchur, Oleksandr, et al. "fev-bench: A Realistic Benchmark for Time Series Forecasting." _arXiv preprint arXiv:2509.26468_ (2025).

**Questions:**

- See Weaknesses.
- I have a question that might be more related to the winner-takes-all paradigm than to your specific method. How do you make sure that the selected best scenario is different so that different weights from the last layer can get a learning signal? Also in datasets where we only observe one realization of the target (and not very similar inputs either), what makes the different scenarios diverse and prevent them from collapsing into the observed sample?
- Can you please elaborate more on the differences between your approach and that of TimeMCL? Skimming through the TimeMCL paper, I can see that the loss formulation has a very similar construction, based on a WTA term and a cross entropy term maximizing the probability of the selected scenario, which to me seems similar if not identic to what's proposed in your paper.

---

> ### Author Response · Authors · 2025-11-21
>
> We thank the reviewer for their detailed feedback and insightful suggestions.
>
> ### **Weakness 1: Expanded Evaluation Scope**
>
> We thank the reviewer for this constructive suggestion. We agree that validating our paradigm on newer, more comprehensive benchmarks is crucial. Following the reviewer's recommendation, we have conducted new experiments on four representative datasets from Gift-eval \[1\] and fev-bench \[2\], selected from four different domains: Hierarchical Sales (retail), M-DENSE (mobility), Hospital Admissions (healthcare), and UCI Air Quality (nature).
>
> The results, presented in the tables below, demonstrate that TimePrism remains highly competitive against SOTA baselines in these new settings, achieving **6 out of 8** SOTA results across the four additional datasets, thereby validating the generalizability of our paradigm.
>
> **Table 1: CRPS on New Benchmarks**
> | Model | UCI | Hosp. | Hier. | M-Den. |
> | :--- | :--- | :--- | :--- | :--- |
> | ETS | 0.450 | 0.585 | 0.990 | 0.782 |
> | Tactis2 | 0.605 | 0.583 | 0.623 | **0.614** |
> | TimeMCL | 0.359 | 0.722 | 1.080 | 0.771 |
> | **TimePrism** | **0.261** | **0.565** | **0.602** | 0.907 |
>
> **Table 2: Distortion on New Benchmarks**
> | Model | UCI | Hosp. | Hier. |M-Den. |
> | :--- | :--- | :--- | :--- | :--- |
> | ETS | 0.811 | 1.300 | 4.030 | 2.230 |
> | Tactis2 | 0.787 | 1.200 |1.580  | 1.130 |
> | TimeMCL | 0.449 | 1.130 | 1.330 | **0.597** |
> | **TimePrism** | **0.394** | **1.060** | **1.030** | 1.110 |
>
> We note that these benchmarks are massive (Gift-eval: 56 datasets, fev-bench: 96 datasets). Due to the limited rebuttal period and computational constraints, we selected four diverse datasets and four representative models: **ETS** (a non-neural numerical baseline), **Tactis-2** and **TimeMCL** (newer and more competitive baselines), and our proposed **TimePrism**. Regarding the M-DENSE dataset, we observed that TimePrism exhibits relatively higher distortion. We hypothesize that the nature of this dataset may be more suitable for RNN backbones, as both Tactis-2 and TimePrism perform suboptimally on this dataset, while TimeMCL remains competitive. This is not a limitation of our new paradigm, but rather a consequence of TimePrism's simple structure. Nevertheless, achieving SOTA results in 15 out of 18 metrics across 9 datasets still demonstrates the effectiveness of the TimePrism model and highlights the potential of the new paradigm.
>
> We believe these extended experiments successfully demonstrate the generalizability and potential of our proposed paradigm. Should this validation be deemed insufficient, we are committed to further expanding the experimental scope in the final version.
>
> [1] Aksu, Taha, et al. "Gift-eval: A benchmark for general time series forecasting model evaluation." arXiv preprint arXiv:2410.10393 (2024).
>
> [2] Shchur, Oleksandr, et al. "fev-bench: A Realistic Benchmark for Time Series Forecasting." arXiv preprint arXiv:2509.26468 (2025).
>
>
> ### **Weakness 2: Validity of Distortion Metric for Tail Events**
>
> We agree with the reviewer that Distortion alone does not prove that our model identifies "low-probability, high-impact" events. We address this point in two ways:
>
> 1. **Qualitative Analysis:** Our qualitative results in **Figure 4** (main paper) provide a direct answer. On the Solar dataset, we identified a "Rare Case" (low peak) that occurred only once among five instances with similar histories. The top panel clearly shows that TimePrism not only generates a scenario that accurately covers this rare event but also correctly assigns it a low probability (represented by the thin, red-tinted line), distinguishing it from the common high-peak futures (which receive high-probability, thick, blue lines).
> 2. **Quantitative Analysis:** We thank the reviewer for the histogram suggestion. However, quantitatively defining "similarity" for both inputs and outputs to build such a histogram is ambiguous. Instead, our **Weighted CRPS** metric serves precisely this quantitative purpose. The CRPS is a strictly proper scoring rule; to achieve a low (good) score, a model *must* assign high probabilities to likely events and low probabilities to rare events. The fact that TimePrism achieves strong results on **both Distortion and CRPS simultaneously** demonstrates that it not only covers the ground truth (low Distortion) but also assigns appropriate probabilities to its scenarios, including those that are rare (low CRPS).

---

> > ### Author Response · Authors · 2025-11-21
> >
> > ### **Weakness 3: Fair Comparison**
> >
> > This is a very insightful point. We respectfully highlight that this seemingly "unfair" comparison is exactly the core issue we aim to investigate in this work.
> >
> > 1. **Implicit Probability in Sampling:** For models capable of modeling a continuous probability density function (PDF), the frequency of sampled trajectories appearing in a specific region naturally correlates with the local PDF value, as in the fundamental principle of Monte Carlo integration. Therefore, treating the obtained samples with equal weights ($1/S$) is statistically reasonable, as the spatial density of the samples themselves implicitly represents the underlying probability distribution.
> >
> > 2. **Scope of Research:** While designing better, model-specific output mechanisms to surpass standard sampling is a critical open question, it would require fundamental architectural changes tailored to each baseline. Such extensive modifications constitute a separate research challenge that is out of the scope of this paper, which focuses on introducing a novel paradigm.
> >
> > 3. **Community Value:** A key contribution of our work is to bring this exact issue to the forefront. By demonstrating the advantages of an explicit {Scenario, Probability} output, we hope to encourage the community to prioritize this problem and explore further solutions beyond the current sampling-dominant approach.
> >
> >
> > ### **Question 1: WTA Scenario Diversity**
> >
> > We thank the reviewer for this important question regarding the stability of the WTA mechanism. This is a well-known challenge often referred to as component or mode collapse.
> >
> > 1. **Preventing Collapse:** Our model, TimePrism, explicitly implements established solutions from the MCL literature to prevent this. As detailed in Appendix C.2.1, we use **Relaxed-WTA** \[3\], which ensures that even *"loser"* scenarios receive a small, non-zero gradient signal (controlled by $\epsilon$). This small update prevents component stagnation ("dead scenarios") and encourages all N\*K scenarios to continuously explore the space. Related work on Annealed-WTA \[4\] further validates this principle by using an annealing schedule to soften assignments, achieving the same goal of ensuring all components are updated. As this is more related to the WTA mechanism than our proposed paradigm and has been well-studied by previous researchers, we placed the details in the appendix rather than providing an extensive analysis in the main text.
> >
> > 2. **Diversity in Absence of Similar Inputs:** The reviewer's second question highlights a key strength of our **combinatorial architecture**. TimePrism does not learn to map one history to one future. Instead, it learns a shared *basis* of M trend and K seasonal components from the *entire* dataset. Even for a unique input, the model's first step is to *compose* N\*K scenarios by combining these general, learned bases (e.g., trend\_m \+ season\_k). The probability layer then assesses the likelihood of each of these *composed* scenarios given the specific input. This compositional structure inherently provides diversity and prevents the model from collapsing to a single output, as it is fundamentally designed to generate a set of futures from shared, learned components rather than memorizing specific input-output pairs.
> >
> >
> > [3] Rupprecht, Christian, et al. "Learning in an Uncertain World: Representing Ambiguity Through Multiple Hypotheses." ICCV 2017.
> >
> > [4] Perera, David, et al. "Annealed Multiple Choice Learning: Overcoming limitations of Winner-takes-all with annealing." NeurIPS 2024.

---

> > > ### Author Response · Authors · 2025-11-21
> > >
> > > ### **Question 2: Difference from TimeMCL**
> > >
> > > This is a key comparison. While both models include WTA-based loss functions, their objectives, architectures, and (most importantly) their probability models are fundamentally different.
> > >
> > > 1. **Core Objective & Architecture:**
> > >    * **TimeMCL:** Is an **auto-regressive RNN** that generates K hypotheses *at each time step*. Its stated goal is to prioritize Distortion (coverage), and the authors note its CRPS (probability) is not SOTA.
> > >    * **TimePrism (Ours):** Is a **non-auto-regressive linear** model that generates N\*K *full scenarios* in one pass. Our goal is to *jointly* solve probability, coverage, and cost.
> > > 2. **Probability Model (The Key Distinction):**
> > >    * **TimeMCL:** Does **not** learn a valid probability distribution. Its "score loss" ($\mathcal{L}^s$, Eq. 5 in \[Cortés et al., 2025\]) is a **BCE loss applied at *each time step***. The final "score" for a trajectory is the **average of these step-wise scores** (Sec 4.2 in \[Cortés et al., 2025\]). This score is not a true probability (it doesn't sum to 1\) and averaging step-wise scores is not probabilistically sound (joint probability requires products). Critically, to compute CRPS, TimeMCL *must still resample* from its K scenarios using these scores as weights (Sec 6.1 in \[Cortés et al., 2025\]), thus failing to solve "Probability Absence" and "Inference Cost".
> > >    * **TimePrism (Ours):** Learns a **true categorical probability distribution**. Our probability loss ($\mathcal{L}_{prob}$, Eq. 10 in our paper) is a **single, global Cross-Entropy loss** over all N\*K *full scenarios*. The probability layer \+ Softmax (Eq. 11\) *guarantees* a valid distribution (sums to 1). We can compute CRPS *directly* using these probabilities, with no resampling.
> > >
> > > **In summary:** TimeMCL's step-wise averaged score is not a true probability. TimePrism learns a more valid probability distribution over complete scenarios, resulting in more competitive CRPS scores. This makes TimePrism the first model to truly realize the {Scenario, Probability} paradigm, solving all three limitations of sampling.

---

> > > > ### Comment · Reviewer_zH2j · 2025-11-27
> > > > **Comment on rebuttal**
> > > >
> > > > I would like to thank the authors for their rebuttal. The raised points about the distortion metric, the fair comparison with other probabilistic baselines have been addressed. The questions about the WTA framework and the comparison with TimeMCL have been answered thoroughly. The raised point about the evaluation scope has been partially addressed. Below are some additional comments:
> > > >
> > > > * I appreciate the additional experiments, however, I still believe that given the simplicity of the method (linear model), a benchmark-level comparison would largely strengthen the paper. This would also have the benefit of not having to run the baselines as you can compare with the results reported in the respective benchmark papers (for gift-eval a leaderboard is also live on huggingface).
> > > >
> > > > * To comment on the histogram idea, a similarity between inputs and outputs is not needed in my opinion. In fact, you could just draw the histogram of the probabilities assigned to the best scenario across samples and datasets. This will show the proportion of events where you correctly identified low-probability scenarios. However, I agree with the authors that Distortion when taken with the CRPS gives the full picture, and with this no additional experiment are needed.
> > > >
> > > > Finally, because most of my concerns were addressed, I'm raising my score to 6.

---

### Author Response · Authors · 2025-12-01
**Summary to AC**

**Summary of Contributions and Reviews**

This work introduces "Probabilistic Scenarios," a paradigm shift from sampling-based forecasting, offering an alternative for reliable decision-making. We are encouraged by the reviewers' recognition of the insight and vision of this work:

* **Well Motivated**: Multiple reviewers praised the clear motivation behind our work. Reviewer zH2j noted the paper is "well motivated," Reviewer VcU5 found the application of discretization "well-motivated," and Reviewer NW9F highlighted that the approach is "well motivated for practical applications."
* **Novelty & Paradigm**: Reviewer zH2j highlights that the paradigm is "innovative" and "opens up a new horizon". Reviewer Ca7z describes it as a "compelling conceptual shift," while Reviewer VcU5 finds the core paradigm "elegantly simple and... highly effective."
* **Performance & Efficiency**: Reviewer Ca7z praises the "competitive accuracy with extreme simplicity" and "strong efficiency advantages." Reviewer NW9F notes the "good trade-off between quality and inference computational cost." Reviewer zH2j confirms the algorithm "performs better than the considered baselines."


**Resolution of Concerns and Rebuttal Updates**

We are pleased to report that the reviewers have reacted positively to our rebuttal. After reviewing our responses to all the reviewers, Reviewer NW9F stated that "I read the other reviews, the results are overall very promising". Furthermore, Reviewer zH2j acknowledged that "most of my concerns were addressed" and consequently **raised their score to 6**, resulting in an **average rating of 6.0**.

To fully address all concerns, we have incorporated the following key updates:
* **Expanded Evaluation Scope**: We conducted experiments on four additional datasets from recent benchmarks (Gift-eval and fev-bench) to demonstrate generalization (Reviewer zH2j).
* **Additional Controlled Ablation**: We implemented TimePrism-iT, adapting the iTransformer architecture to our paradigm. This demonstrated that the performance gains stem from the Probabilistic Scenarios paradigm itself, not just the linear backbone (Reviewer VcU5).
* **Direct Comparison**: We provided results with $N=16$ scenarios to ensure a direct comparison with TimeMCL, showing TimePrism remains superior even under restricted constraints (Reviewer NW9F).
* **Calibration Diagnostics**: We added Reliability Diagrams and PIT histograms, demonstrating strong probabilistic calibration (Reviewer Ca7z).

---

### Meta-Review · Area_Chair_w2Li · 2026-01-04

**Summary:**

The paper proposes as simple yet accurate approach for probabilistic forecasting that competes with much more complex methods. During the reviews, the authors raised some concerns about the empirical evaluation (including metrics, calibration and additional datasets), conceptual comparison with existing methods and extensions. I believe that the authors did a great job during the rebuttal. As a consequence, while I agree with Reviewer zH2j that a "a benchmark-level comparison would largely strengthen the paper, I still believe that the paper is above the acceptance bar. Thus, I encourage the authors to revise their paper to include the additional results and discussion in the camera ready. Further benchmarking will only improve the potential impact of their paper, hence,  if time permits, they shoudl consider further expanding the results for the final version.

**Reviewer Concerns:**

I believe that the main reviewers' concerns have been successfully addressed and no major issues on the paper remain.

**Reviewer Scores:**

After rebuttal, all but one reviewer acknowledged their positive opinion on the paper. I believe that the authors properly responded also to Reviewer VcU5. So I would have not been surprised by an increase on their score.

---

### Decision · Program_Chairs · 2026-01-26

Accept (Poster)